# Integrating multimodal and multiscale connectivity blueprints of the human cerebral cortex in health and disease

**Justine Y. Hansen**[1], **Golia Shafiei**[2], **Katharina Voigt**[3,4], **Emma X. Liang**[4], **Sylvia M. L. Cox**[5], **Marco Leyton**[1,5], **Sharna D. Jamadar**[3,4], **Bratislav Misic**[1] *

**1** Montréal Neurological Institute, McGill University, Montréal, Canada, **2** Department of Psychiatry, Perelman School of Medicine, University of Pennsylvania, Philadelphia, Pennsylvania, United States of America, **3** Turner Institute for Brain and Mental Health, Monash University, Clayton, Australia, **4** Monash Biomedical Imaging, Monash University, Clayton, Australia, **5** Department of Psychiatry, McGill University, Montréal, Canada

* bratislav.misic@mcgill.ca

**Data Availability Statement:** All code and data used to perform the analyses are available on

## Abstract

The brain is composed of disparate neural populations that communicate and interact with one another. Although fiber bundles, similarities in molecular architecture, and synchronized neural activity all reflect how brain regions potentially interact with one another, a comprehensive study of how all these interregional relationships jointly reflect brain structure and function remains missing. Here, we systematically integrate 7 multimodal, multiscale types of interregional similarity ("connectivity modes") derived from gene expression, neurotransmitter receptor density, cellular morphology, glucose metabolism, haemodynamic activity, and electrophysiology in humans. We first show that for all connectivity modes, feature similarity decreases with distance and increases when regions are structurally connected. Next, we show that connectivity modes exhibit unique and diverse connection patterns, hub profiles, spatial gradients, and modular organization. Throughout, we observe a consistent primacy of molecular connectivity modes—namely correlated gene expression and receptor similarity—that map onto multiple phenomena, including the rich club and patterns of abnormal cortical thickness across 13 neurological, psychiatric, and neurodevelopmental disorders. Finally, to construct a single multimodal wiring map of the human cortex, we fuse all 7 connectivity modes and show that the fused network maps onto major organizational features of the cortex including structural connectivity, intrinsic functional networks, and cytoarchitectonic classes. Altogether, this work contributes to the integrative study of interregional relationships in the human cerebral cortex.

## Introduction

Brain connectivity classically refers to the physical neural fibers that link disparate neuronal populations. Axonal projections can be reconstructed by imaging fluorescently labeled proteins that are either injected into or genetically expressed by a cell, or by stacking electron

GitHub at https://github.com/netneurolab/hansen_
many_networks/tree/v1.0.0 and on Zenodo at
https://zenodo.org/record/8250809 (DOI: 10.5281/
zenodo.8250809).

**Funding:** BM acknowledges support from the
Natural Sciences and Engineering Research
Council of Canada (NSERC), Canadian Institutes of
Health Research (CIHR), Brain Canada Foundation
Future Leaders Fund, the Canada Research Chairs
Program, the Michael J. Fox Foundation, and the
Healthy Brains for Healthy Lives initiative. JYH
acknowledges support from the Helmholtz
International BigBrain Analytics & Learning
Laboratory, the Natural Sciences and Engineering
Research Council of Canada, and The Neuro Irv
and Helga Cooper Foundation. SDJ acknowledges
support from the National Health and Medical
Research Council of Australia (APP1174164). The
funders had no role in study design, data collection
and analysis, decision to publish, or preparation of
the manuscript.

**Competing interests:** The authors have declared
that no competing interests exist.

**Abbreviations:** ADHD, attention-deficit/
hyperactivity disorder; AHBA, Allen Human Brain
Atlas; ANTs, Advanced Normalization Tools; ASD,
autism spectrum disorder; DWI, diffusion weighted
imaging; EEG, electroencephalography; ENIGMA,
Enhancing Neuroimaging Genetics through Meta-
Analysis; FDG-PET, fluorodeoxyglucose positron
emission tomography; fMRI, functional magnetic
resonance imaging; HCP, Human Connectome
Project; LCMV, linearly constrained minimum
variance; MDD, major depressive disorder; MEG,
magnetoencephalography; MRI, magnetic
resonance imaging; MT, magnetization transfer;
OCD, obsessive-compulsive disorder; SNF,
similarity network fusion; SSP, Signal-Space
Projection.

microscopy images of thinly sliced brain sections [1,2]. At the macroscale, diffusion-weighted imaging can be used to trace large fiber bundles that connect pairs of brain regions in vivo, which collectively constitute the structural connectome [3,4]. Across organisms, spatial scales, and imaging techniques, the brain's white-matter architecture ("structure") exhibits hallmark features including a prevalence of short range connections resulting in functionally segregated modules [5,6], and a small number of disproportionately densely interconnected hubs [7]. Ultimately, studying the brain's structural connectome has advanced our understanding of how information is transmitted [8,9], how brain structure supports function [10], and how perturbations may result in network-defined pathology spread [11].

However, the graph representation of the structural connectome, in which regional nodes are identical, does not account for the molecular and physiological heterogeneity that exists in the brain. An emerging representation of connectivity is feature similarity: If 2 brain regions exhibit similar biological features, we might expect them to be related to one another and engaged in common function [12–16]. Neuroanatomical tract-tracing studies in nonhuman primates have extensively shown that biological feature similarity is fundamental to brain organization [17,18]. These pioneering studies demonstrated that neuronal projection patterns can be predicted based on the laminar differentiation of the source and target regions [19] and has been extended to human prefrontal cortex and other model organisms [20,21]. Further-more, local differences in laminar architecture follow a gradient of receptor density [22,23] and synaptic plasticity [24], indicating an alignment between multiple local features and con-nectivity. However, these studies are currently limited to qualitative measurements of cytoarchitectonic similarity, small subsets of brain regions, model organisms, and to a single perspective of molecular make-up (but see [25]).

An alternative approach is to acquire densely sampled whole-cortex neuroimaging data across large populations with the goal of constructing a connectivity matrix based on feature similarity. This approach is already widely used on the BOLD signal where haemodynamic time courses are correlated with each other and also exists for time series measures from other imaging modalities such as magneto-/electroencephalography (MEG/EEG) and dynamic FDG-fPET (all called "functional connectivity") [26–30]. In cases where multiple measures of a feature exist at each brain region, such as gene expression levels across many genes, interre-gional similarity can be estimated with respect to a single local feature [14,23,31–35]. In each case, the ensuing region × region correlation matrix represents a form of connectivity between brain regions.

As multiple estimates of interregional similarity become available through emerging tech-nologies and data sharing efforts, it becomes possible to integrate them into a single framework and deduce how they interact with one another, and in what ways they are unique or comple-mentary. For example, cortical structure is heterogeneously coupled to haemodynamic func-tional connectivity along the sensory-association cortical hierarchy [36,37]. Information about interregional feature similarity adds additional insight on how structure supports function and has been shown to improve the structure-function concordance [23,38,39]. The advance in neuroimaging techniques and data sharing standards has now made it possible to study multi-ple forms of interregional relationships jointly, spanning a range of spatial and temporal scales. The future of connectomics is therefore no longer limited to structural connectivity, but can be approached from a multimodal, multiscale angle.

Here, we integrate 7 layers of interregional relationships, including gene expression, recep-tor density, cellular composition, metabolism, electrophysiology, and temporal fingerprints, to assemble a comprehensive, multiscale wiring blueprint of the cerebral cortex. Although they are all effectively networks reconstructed by correlating feature similarity, hereafter, we refer to them as connectivity modes. First, we establish the common and unique manners in which

connectivity modes reflect cortical structure and geometry. Next, we identify cross-modal hubs as well as circuits that consistently display large interregional similarity across multiple connectivity modes. We then test how different connectivity modes capture patterns of abnormal cortical thickness across 13 neurological, psychiatric, and neurodevelopmental disorders. Moreover, we show that connectivity modes demonstrate diverse gradient and modular decompositions. Finally, we iteratively fuse all 7 connectivity modes into a single multimodal network. All 7 connectivity modes are publicly available in 3 parcellation resolutions (https://github.com/netneurolab/hansen_many_networks), in hopes of facilitating integrative, multiscale analysis of human cortical connectivity.

## Results

For each brain feature, a similarity network can be represented as a region × region matrix. Rows and columns represent cortical regions, and elements—the edges of the similarity network—represent how similarly 2 regions present the specific feature. This similarity can also be thought of as connectedness, such that 2 regions that share similar features are considered strongly connected. For simplicity, we therefore refer to correlation-based similarity as "connectivity" and the similarity networks as "connectivity modes." To comprehensively benchmark cortical connectivity modes, we construct and analyze 7 different connectivity matrices, spanning multiple spatial and temporal scales. These include: (1) correlated gene expression, describing transcriptional similarity across >8,000 genes from the Allen Human Brain Atlas (AHBA) [40]; (2) receptor similarity, describing how correlated pairs of cortical regions are in terms of protein density of 18 neurotransmitter receptors/transporters [23]; (3) laminar similarity, describing how correlated pairs of cortical regions are in terms of cell-staining intensity profiles from the BigBrain atlas [14,41]; (4) metabolic connectivity, measured as the correlation of dynamic FDG-PET (glucose uptake) time series [29,42]; (5) haemodynamic resting-state connectivity, measured as the correlation of functional magnetic resonance imaging (fMRI) BOLD time series from the Human Connectome Project (HCP) [43]; (6) electrophysiological connectivity, measured as the first principal component of resting-state magnetoencephalography (MEG) connectivity across 6 canonical frequency bands from the HCP [43,44]; and (7) temporal profile similarity, a comprehensive account for dynamic similarity (above and beyond a Pearson's correlation between time series, as is the case in haemodynaimc connectivity) which is measured as the correlation between time series features of the fMRI BOLD signal [45–47]. To facilitate comparison between networks, and to mitigate differences between data types and processing pipelines, each network was parcellated to 400 cortical regions and edge values were normalized using Fisher's $r$-to-$z$ transform [48]. Networks were also parcellated to an alternative functional and anatomical cortical atlas in multiple resolutions (100 and 68 cortical regions) for the sensitivity and replication analyses (see Sensitivity and replication analysis).

### Common organizational patterns of connectivity modes

In Fig 1A, we visualize each normalized connectivity matrix as a heatmap where the colorbar limits are −3 and 3 standard deviations of the edge weight distribution (for edge weight distributions, see S1A Fig). Cortical regions are ordered by left then right hemisphere. Within each hemisphere, regions are further stratified by their membership in the 7 canonical intrinsic functional networks (Schaefer-400 parcellation [48,49]). Homotopic connections stand out, indicating that homologous cortical regions in left and right hemispheres are consistently similar to each other no matter the biological feature (S1B Fig [50,51]). Previous work has hypothesized that cortical dynamics in homotopic regions are synchronized due to common

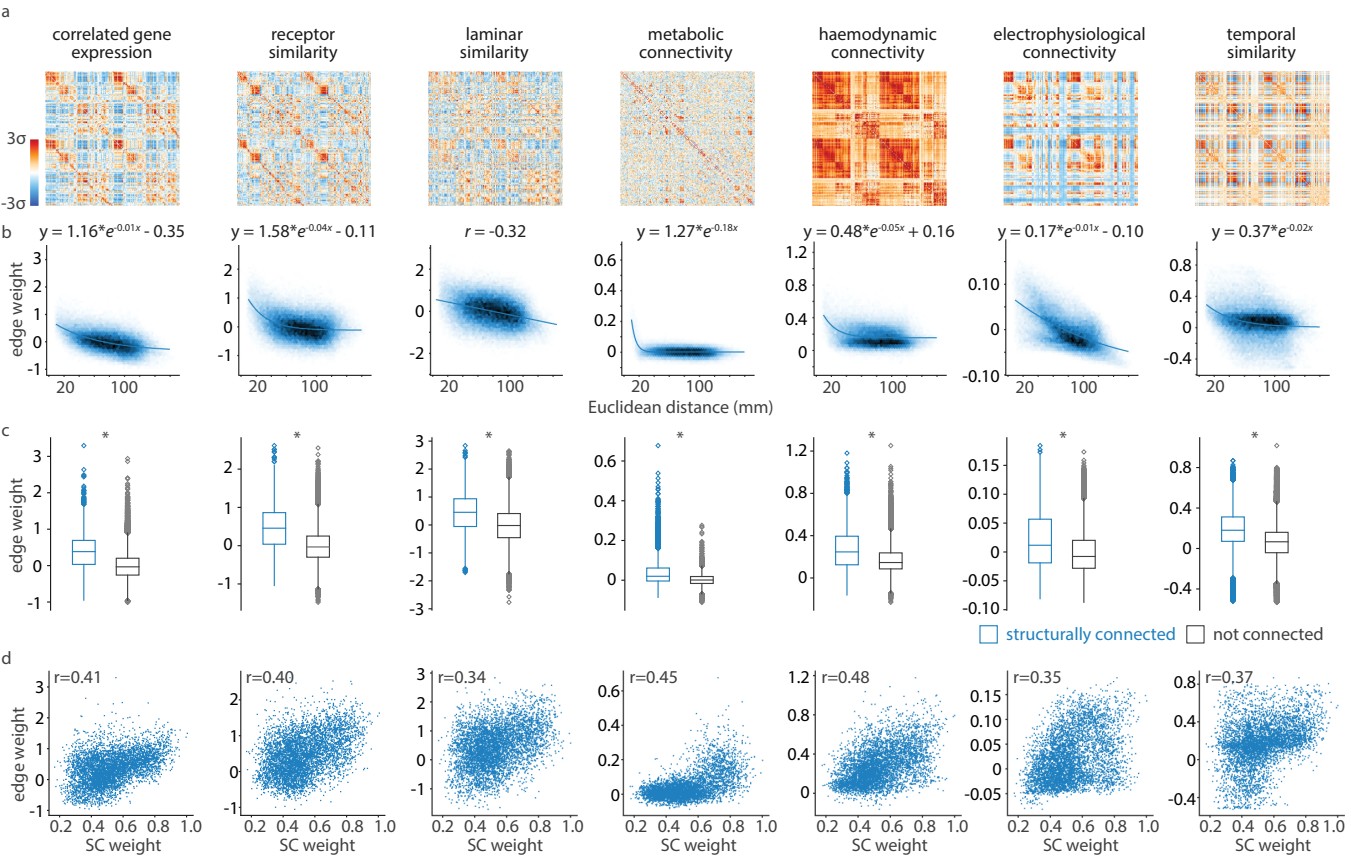

**Fig 1. Common organizational patterns of connectivity modes.** Each connectivity mode is represented as a normalized similarity *matrix*, where elements of the matrix index how similarly two cortical regions present a specific feature. (a) Connectivity modes are shown as heatmaps, ordered according to the 400-region Schaefer parcellation [48]. The colorbar limits are −3 to 3 standard deviations of the edge weight distribution. (b) Edge weights between every pair of cortical regions (i.e., upper triangular elements) decrease with Euclidean distance across all 7 connectivity modes. Darker color represents greater density of points. Exponential equations or Spearman correlation coefficients are shown depending on whether the relationship is better fit by an exponential or linear function. Similar relationships with geodesic distance are shown in S2 Fig. (c) Edge weight distributions are visualized separately for edges that also exist in the structural connectome (blue) and those that do not (gray), according to a group-consensus structural connectome from the HCP [43]. Structurally connected cortical regions show greater similarity than regions that are not structurally connected. Boxplots represent the first, second (median), and third quartiles, whiskers represent the non-outlier end-points of the distribution, and diamonds represent outliers (>1.5 inter-quartile range). (d) For edges that also exist in the structural connectome, connectivity mode edge weight increases with the strength of the structural connection. The data underlying this figure can be found at https://github.com/netneurolab/hansen_many_networks. HCP, Human Connectome Project.

brainstem input [52,53]; our work opens an additional hypothesis that similarities in dynamics may also be related to similar molecular composition.

Visually, each connectivity mode demonstrates nonrandom network organization, which we explore in subsequent sections. Furthermore, similarity between cortical regions decreases as both Euclidean and geodesic distance between cortical regions increases (Figs 1B and S2), consistent with the notion that proximal neural elements are more similar to one another [18,23,31,45,54,55]. However, there is variability in how feature similarity decreases with distance. For example, dynamic modes demonstrate stronger exponential relationships, whereas molecular modes demonstrate either weak exponential or linear (in the case of laminar similarity) fits.

We next sought to relate each connectivity mode to the brain's underlying structural architecture. We constructed a weighted structural connectome using diffusion-weighted MRI data from the HCP; this network represents whether, and how much, 2 cortical regions are

connected by white matter streamlines. We find that, across all 7 connectivity modes, cortical regions that are physically connected by white matter show greater feature similarity than those that are not connected, suggesting that biologically similar neuronal populations are in direct communication (Fig 1C). These differences are greater than in a population of degree- and edge length-preserving surrogate structural connectomes, indicating that the effect is specifically due to wiring rather than spatial proximity [56]. Notably, neuroanatomical studies in model organisms have found that cytoarchitectonic similarity predicts neuronal projections better than distance [57–59]; we expand on this by showing that all connectivity modes demonstrate greater similarity for structurally connected cortical regions in the human. Finally, for the subset of edges with a structural connection, we find a correlation between the strength of the structural connection and each connectivity mode's edge weight (Fig 1D) [59,60]. Altogether, we find that connectivity modes demonstrate commonalities that respect distance, neuroanatomy, and anatomical connectivity, regardless of imaging modality or biological mechanism.

## Structural and geometric features of connectivity modes

Although connectivity modes share organizational properties, the median correlation between them is $r = 0.25$ (range: $r = 0.10–0.53$; S3 Fig). In other words, connectivity modes are not redundant. To directly compare edge weights across connectivity modes, we converted edge weights to ranks, such that the smallest (i.e., most negative) edge is ranked 1 and the strongest (i.e., most positive) edge is ranked 79,800 (equal to the number of edges in each network, under the 400-region Schaefer parcellation). We focus on 2 metrics to classify edges between cortical regions: distance (brain geometry) and structural connectivity (brain structure).

Spatial proximity influences interregional similarity, such that proximal regions tend to share similar biological and physiological features (Fig 1B) [56,59,61]. We therefore sought to investigate how distance shapes interregional feature similarity in greater detail and in a comparative manner. We first bin all 79,800 edges into 50 equally sized bins (1,596 edges per bin). For each connectivity mode separately, we calculate the median edge rank within each bin (Fig 2A). Median edge rank decreases as the distance between cortical regions in each bin increases, consistent with our finding in Fig 1B. We find 2 broad patterns: receptor similarity, temporal similarity, haemodynamic connectivity, and metabolic connectivity show moderate decrease of edge strength with distance, whereas correlated gene expression, laminar similarity, and especially electrophysiological connectivity demonstrate a sharper decrease of edge strength with distance. In other words, distance plays a unique role in shaping each individual connectivity mode, with electrophysiological connectivity, laminar similarity, and correlated gene expression being most influenced by distance. That receptor similarity is grouped with predominantly dynamic modes (haemodynamic, metabolic, and temporal similarity) may reflect the influence that receptor density has on cortical dynamics.

We next shift our focus to the subset of edges with an anatomical connection, according to the structural connectome ($N = 4,954$ out of 79,800 edges). For each connectivity mode, we plot the distribution of rank-transformed feature similarity (edge rank) for these edges that also exist in the structural connectome (Fig 2B). This lets us determine which connectivity modes demonstrate the greatest coupling between high interregional feature similarity and structural connectivity: namely, receptor similarity and correlated gene expression. Previous work has found a close correspondence between cytoarchitecture and neuronal projections in macaque brains [18,62]; our findings suggest a possible genetic and neuroreceptor mechanism underlying this relationship. The primacy of molecular connectivity modes is a finding that returns in the next analysis and when we compare connectivity modes to disease pathology (Connectivity modes and disease-specific abnormal cortical thickness).

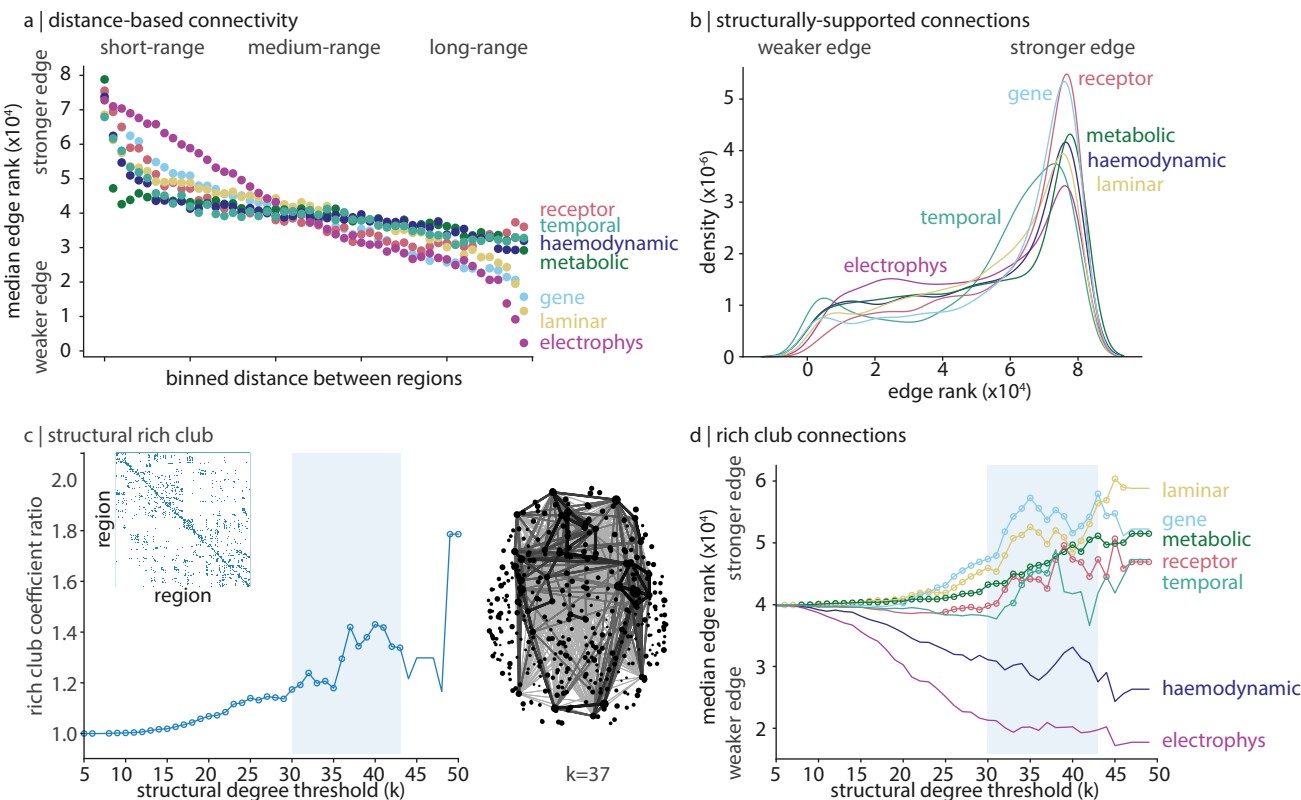

**Fig 2. Structural and geometric features of connectivity modes.** To compare edge weights across networks, edges are rank-transformed. (a) Edges are binned into 50 equally sized bins of increasing Euclidean distance (79,800 edges total under the 400-region Schaefer parcellation, 1,596 edges per bin). For each connectivity mode, the median edge rank is plotted within each bin. (b) A kernel density estimation is applied on the rank-transformed feature similarity (edge rank) distribution of edges that also have a structural connection, for each connectivity mode. (c) For a structural degree threshold k ∈ [5, 50], we calculate the rich club coefficient ratio and show a characteristic increase in rich club coefficient ratio when 30 ≤ k ≤ 43. Circles indicate structural degree thresholds where the rich club coefficient ratio is significantly greater than a null distribution of ratios calculated using a degree-preserving rewired network (1,000 repetitions). On the right, we show the set of structural edges connecting regions with structural degree ≥37. Edge shade and thickness are proportional to edge weight, and point size is proportional to structural degree. The binary structural connectome is shown in the inset. (d) For each k ∈ [5, 50] and for each connectivity mode, we calculate the median edge rank of structurally-supported edges that connected regions with structural degree ≥ k. Circles indicate structural degree thresholds where the median rich-link edge rank of a connectivity mode is significantly greater than the edge rank of all other structurally supported edges (Welch's t test, one-sided). The data underlying this figure can be found at https://github.com/netneurolab/hansen_many_networks.

We next track how edge strength changes depending on the structural embedding of each cortical region. We focus on the cortex's rich club: a set of disproportionately interconnected high-degree regions that is thought to mediate long-range information propagation and integration [7,63]. Is this rich club architecture supported by specific biological and physiological features? To address this question, for each structural degree threshold k∈[5,50] (where structural degree is defined as the number of structural connections made by a cortical region), we calculate the rich club coefficient ratio on the binary structural connectome: the tendency for regions of degree ≥k to be preferentially connected to one another, with respect to a population of degree-preserving surrogate networks. We find that the rich club coefficient ratio is inflated at approximately 30≤k≤43, confirming the existence of rich club organization (Fig 2C). This topological rich club regime denotes a degree range where cortical regions are unexpectedly densely interconnected [64]. Next, for each connectivity mode at each k, we calculate the median edge rank of all structurally supported edges that link 2 cortical regions with degree ≥k (Fig 2D). Moreover, we ask whether within-set edge ranks (i.e., edges connecting regions with degree ≥k) are statistically greater than all other edges (Welch's one-sampled t test).

We find that edges in the cortex's topological rich club regime are particularly dominated by molecular features (e.g., laminar similarity, correlated gene expression, and receptor similarity) [57]. Haemodynamic and electrophysiological connectivity are especially weak for links between high-degree regions, and temporal similarity is unstable. Metabolic connectivity is an additional connectivity mode that demonstrates significantly increased edge strength for links between high-degree regions, suggesting that energy consumption is synchronized between structural hubs [63,65–67]. Collectively, these findings indicate that the rich club may reflect coordinated patterns of interregional microscale similarity across multiple molecular features. On the other hand, the rich club is not characterized by similar neural dynamics, possibly related to the functional flexibility of these regions [68].

## Cross-modal hubs

Mapping hubs in the human brain has been a topic of great interest in the last 15 years, but the majority of our knowledge comes from anatomical and haemodynamic connectivity [69,70]. For a more comprehensive understanding of brain regions that make many strong connections, it would be important to map their connectivity profiles at different levels of organization. We therefore ask whether there exist edges that are consistently high-strength, and if so, which cortical regions, which we call cross-modal hubs, make these connections. For every connectivity mode, we show an axial view of the 0.5% strongest edges (Fig 3A; see S4 Fig for coronal and sagittal views). Interestingly, high-strength edges vary across connectivity modes: some networks form densely interconnected cores (i.e., electrophysiological connectivity and temporal similarity), some emphasize long-range (i.e., haemodynamic connectivity) or short-range (i.e., metabolic connectivity) connections, and others appear more nonspecific (i.e., correlated gene expression, receptor similarity, and laminar similarity; Fig 3A). This variability is also reflected in the hubness profiles of each connectivity modality, where a region's hubness is defined as the sum of the rank-transformed edge weights between it and all other regions (Fig 3B). The variability of hubness points to the importance of characterizing network architecture from multiple complementary perspectives.

Are there consistencies in high-strength edges and regions? Previous work has shown that the cortex can be organized into modules of regions that are either functionally similar ("intrinsic networks" [49]) or cellularly similar ("cytoarchitectonic classes" [71,72]). We wanted to know whether connectivity modes across multiple scales emphasize edges that link cortical regions within these functional and cytoarchitectonic networks, regardless of whether the connectivity mode represents cortical function or cellular composition. For a given network classification (e.g., intrinsic networks), we call edges that join 2 cortical regions in the same network (e.g., the visual network) intra-class edges [32]. We then calculate how many of the $x$ strongest edges in a given connectivity mode overlap with intra-class edges. We let $x$ vary in increments of 0.5% from 0.5% to 5% of the strongest edges in a connectivity mode.

For intrinsic networks (Fig 3C, left), the strongest edges in the haemodynamic network are almost entirely intra-class edges (90.2% for the top 0.5% strongest edges, and 72.2% for the top 5% edges). The strongest edges in correlated gene expression are also primarily intra-class edges (88.7% for the top 0.5% strongest edges) but this ratio decreases to 52.8% at 5% of the strongest edges. Meanwhile, for cytoarchitectonic classes (Fig 3C, right), receptor similarity, correlated gene expression, and metabolic connectivity most maximize intra-class edges. Across both intrinsic and cytoarchitectonic networks, temporal similarity retains the fewest intra-class edges. Nonetheless, the negative slopes in Fig 3C indicates that, for every connectivity mode, strongest edges are preferentially edges that connect cortical regions within the same functional and cytoarchitectonic network (Fig 3C). More generally, when we consider the

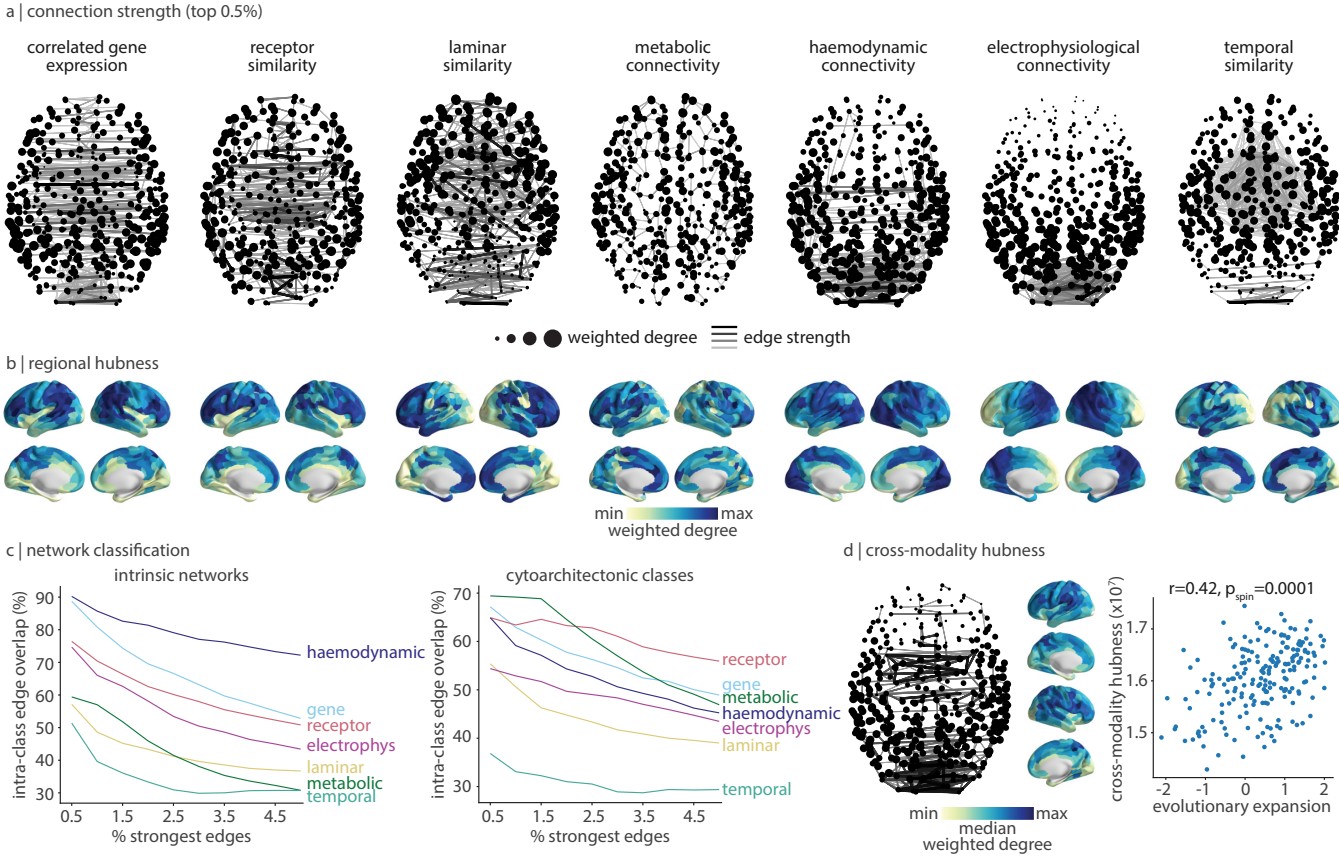

**Fig 3. Cross-modal hubs.** (a) For each connectivity mode, we plot the 0.5% strongest edges. Darker and thicker edges indicate stronger edges. Points represent cortical regions and are sized according to the sum of edge weights (weighted degree). Cortical views are axial, with anterior regions at the top of the page (for coronal and sagittal views, see S4 Fig). (b) For each connectivity mode, regional hubness is defined as the sum of rank transformed edge weights across regions. (c) For a varying threshold of strongest edges (0.5%–5% in 0.5% intervals), we calculate the proportion of edges that connect 2 regions within the same intrinsic network [49] (left) and cytoarchitectonic class [71] (right). (d) Across all 7 connectivity modes, we calculate the median edge rank of each edge and plot the 0.5% strongest edges (left). Likewise, we calculate the median hubness (shown in panel b), which we find is significantly correlated with evolutionary cortical expansion (r = 0.42, $p_{spin}$ = 0.0001) [73]. The data underlying this figure can be found at https://github.com/netneurolab/hansen_many_networks.

median edge rank across all connectivity modes, we find that consistently high-strength edges primarily connect visual, posterior parietal, and anterior temporal regions (Fig 3D, left).

Finally, we focus on the cortical regions: given the spatial diversity of hub profiles (Fig 3B), are there regions that consistently show relatively high weighted degree—that is, are consistently similar to other brain regions—across multiple connectivity modes? We quantify cross-modal hubness as the median hubness across connectivity modes (i.e., the median across brain plots shown in Fig 3B). We find that transmodal eulaminate regions such as the supramarginal gyrus, superior parietal cortex, precuneus, and dorsolateral prefrontal cortex are most consistently similar to other cortical regions across 7 biological phenotypes, from molecular composition to neural dynamics (Fig 3D, right). Interestingly, the regions identified as cross-modal hubs are commonly thought of as hubs in the structural connectome; we find that they also demonstrate large feature similarity across multiple levels of organization.

Why are some cortical regions highly similar to many other regions across multiple spatial scales and biological mechanisms? We hypothesized that cross-modal hubs are more cognitively flexible and able to support higher-order, evolutionarily advanced cognitive processes. We therefore correlated cross-modal hubness with a map of evolutionary cortical expansion

[73]. Indeed, the identified cross-modal core coincides with cortical regions that are more expanded across phylogeny ($r = 0.43$, $p_{spin} = 0.0001$). In other words, cortical regions that are expanded in humans and therefore likely involved in higher-order cognition share many features across multiple scales, suggesting they can integrate signals from a more diverse set of neural circuits. Ultimately, hubs that are defined using connectivity modes other than the classical structural connectome provide novel perspectives on how regions participate in neural circuits.

## Connectivity modes and disease-specific abnormal cortical thickness

Pioneering studies in postmortem tissue gave rise to the theory that the physicochemical composition of neurons at local brain regions results in a selective vulnerability to brain disease [74]. Other classical studies have shown that disease propagation in the cerebral cortex is related to microscale features such as myelination [75]. These propagation patterns have been successfully modeled at the level of the whole-cortex using the structural connectome, and often perform better when informed by local biological features such as the expression of a specific gene [76,77]. Recent findings build on this notion and posit that the course and expression of multiple brain diseases is mediated by shared molecular vulnerability rather than a single molecular perturbation [33,78]. We therefore tested whether disease propagation patterns derived from the connectivity modes could predict abnormal cortical thickness patterns for 13 different neurological, psychiatric, and neurodevelopmental diseases and disorders from the Enhancing Neuroimaging Genetics through Meta-Analysis (ENIGMA) consortium ($N = 21,000$ patients, $N = 26,000$ controls) [33,79,80]. The disease-specific abnormal cortical thickness patterns are regional z-scored case-versus-control effect sizes, representing deviation from normative cortical thickness. We refer to these regional values as "abnormal cortical thickness" or simply "abnormality."

We define the "exposure" that region $i$ has to region $j$'s pathology as the product between the $(i,j)$-edge strength ($c_{ij}$ if $c_{ij}>0$) and region $j$'s abnormal cortical thickness ($d_j$) (Fig 4A) [33,81–83]. Then, the global disease exposure to region $i$ is the mean exposure between region $i$ and all other regions in the network with positive edge strength (note that we find consistent results when we use all edges of the network (S5a Fig)). Finally, we correlate abnormal cortical thickness with disease exposure to determine whether the disease demonstrates a cortical disease profile that reflects the underlying connectivity mode (Fig 4A, right). Given a disease where greater disease exposure results in greater abnormal cortical thickness, we would expect to find a large positive correlation. This analysis is repeated for each connectivity mode and each disorder, and correlation coefficients are visualized in Fig 4B (see S6 Fig for results including the fused network (Fusing connectivity modes)).

We find that correlated gene expression and receptor similarity most consistently amplify the exposure of pathology in a manner that closely resembles the structural cortical profile of the disease. Interestingly, when we repeat the analysis using only negative edges ($c_{ij}$ if $c_{ij}<0$), we find opposite results: Cortical regions with high abnormality are negatively connected (that is, are dissimilar to) regions with low abnormality (S5B Fig). This suggests that dissimilarity may attenuate disease spread. By repeating the analysis using weighted structural connectivity (in which case, we only consider structurally connected regions) and Euclidean distance between cortical regions (in which case, we always consider the full network), we are able to uncover cases where feature similarity amplifies disease exposure more than structure or distance alone (Fig 4C). Abnormal cortical thickness patterns of psychiatric disorders in particular (e.g., MDD, schizophrenia, bipolar disorder, OCD) are better explained by correlated gene expression and receptor similarity than structure or distance. This integrative analysis makes it

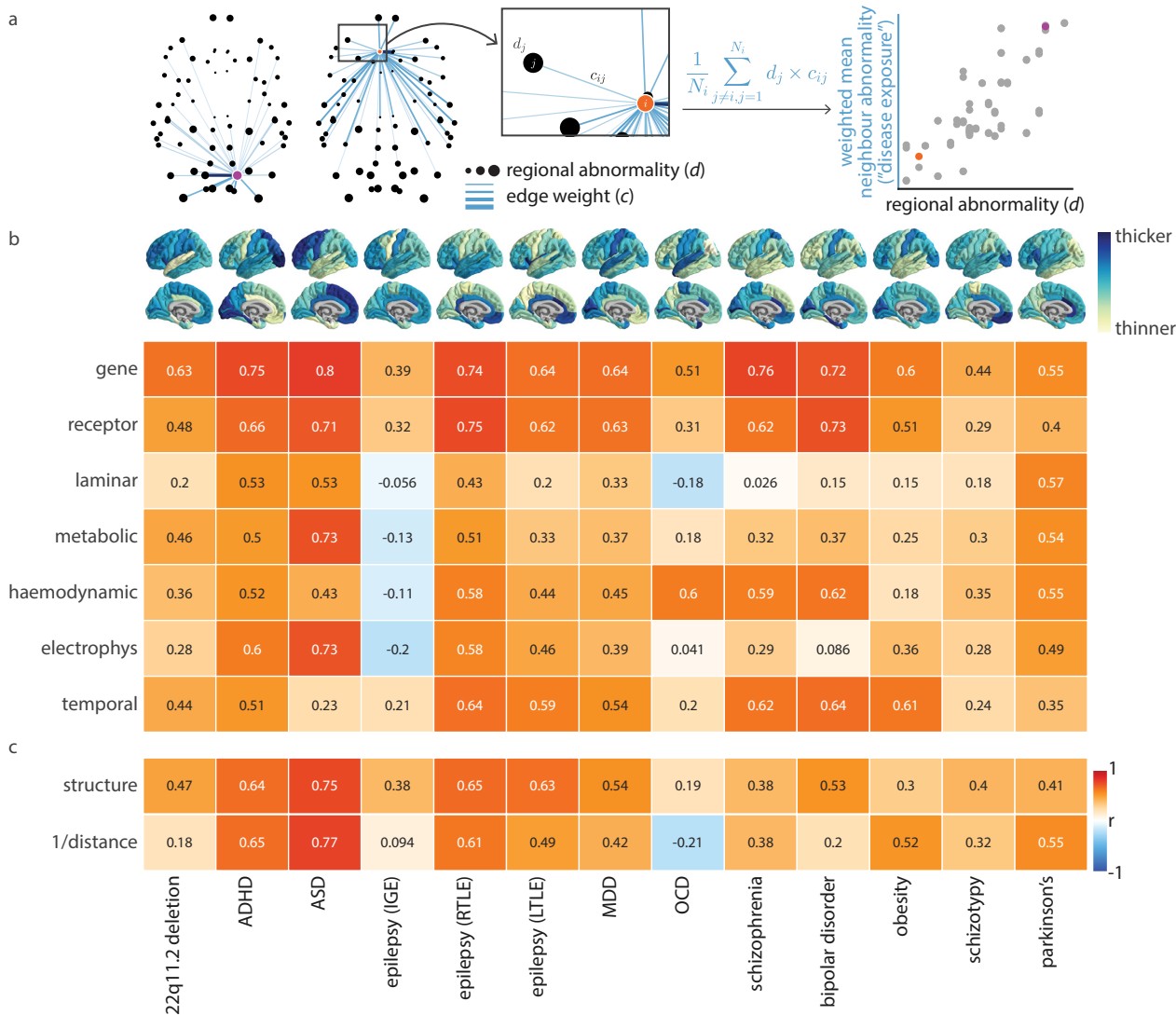

**Fig 4. Contributions of connectivity modes to disease vulnerability.** Abnormal cortical thickness profiles for 13 neurological, psychiatric, and neurodevelopmental disorders were collected from the ENIGMA consortium (cortex plots shown in panel b; $N = 21,000$ patients, $N = 26,000$ controls [79,80]). (a) Given a specific disorder and connectivity mode, $d_j$ represents the abnormal cortical thickness of region $j$, and $c_{ij}$ represents the edge weight (similarity) between regions $i$ and $j$. For every region $i$, we calculate the average abnormal cortical thickness of all other regions $j \neq i$ in the network, weighted by the edge strength ("disease exposure"; note that we omit negative connections, such that $N_i$ represents the number of positive connections made by region $i$). Next, we correlate disease exposure and regional abnormal cortical thickness across cortical regions (scatter plot; points represent cortical regions). We show the connectivity profiles of 2 example regions (highlighted in purple in the left brain network and orange in the right brain network). (b) The analytic workflow presented in panel (a) is repeated for each disorder and connectivity mode, and we visualize Spearman correlations in a heatmap. (c) This analysis is repeated for weighted structural connectivity (where we only consider structurally connected regions), and Euclidean distance (where we always consider all regions in the network). We also repeat this analysis for the fused network (see Fusing connectivity modes), and results are shown in S6 Fig. The data underlying this figure can be found at https://github.com/netneurolab/hansen_many_networks. ADHD, attention-deficit/hyperactivity disorder; ASD, autism spectrum disorder; ENIGMA, Enhancing Neuroimaging Genetics through Meta-Analysis; MDD, major depressive disorder; OCD, obsessive-compulsive disorder.

possible to hone in on the imaging modalities and biological mechanisms that might most reflect cortical pathology in a disease-general manner. Furthermore, it demonstrates the value in employing feature similarity as a network rather than limiting network models to the structural connectome.

## Gradients and modules of connectivity modes

We next consider how each connectivity mode is intrinsically organized, both in terms of axes of variation (i.e., spatial gradients) and network modules [14,22,84–86]. The principal gradient, quantified as the first principal component of a connectivity mode, is a regional quantification of how feature similarity varies across the cerebral cortex. They can be interpreted as a single-dimensional representation of the connectivity mode and will highlight the regions that are especially similar to or dissimilar from one another. We start by studying an underappreciated element of the principal component: How much variance is explained by (i.e., how representative is) each principal gradient? We find that the prominence of the first gradient can vary substantially across connectivity modes (Fig 5A). For example, the temporal similarity gradient is especially dominant (accounting for 73.8% of variance), while the metabolic connectivity gradient is especially nondominant (accounting for 12.7% of variance; Fig 5B). Furthermore, we find that cortical gradients do not all follow a uniform sensory-association axis [87–89], rather, the first principal component of each connectivity mode varies considerably (median absolute correlation between gradients $r$ = 0.36; Fig 5C).

An alternative perspective of intrinsic network organization comes from considering whether and how the network clusters into segregated modules [5]. In other words, which subsets of cortical regions are similar to one another (according to a specific connectivity mode) and are these modules consistent across connectivity modes? We apply the Louvain community detection algorithm to each connectivity mode to search for groups of regions that exhibit high within-module similarity and high between-module dissimilarity [90,91]. The Louvain algorithm is unsupervised and does not require a predefined number of clusters as input; instead, the resolution parameter ($\gamma$) tunes the ease with which more communities are detected (larger $\gamma$ results in more communities being identified). To get a sense of the resolution of each network (i.e., the number of communities the network might naturally exhibit, if at all), we track the number of communities identified by the Louvain community detection algorithm across different values of $\gamma$ (Fig 5D). We find that the community detection solution for electrophysiology is highly unstable, with the number of identified communities changing rapidly with small changes in $\gamma$. The most stable solution at $\gamma$ = 1 simply delineates the main cortical lobes that suggests that electrophysiological connectivity organization is better described as a gradient but not as distinct modules of brain regions. Haemodynamic connectivity and temporal similarity show a similar trend, where partitions of greater than approximately 5 networks become increasingly unstable. Meanwhile, correlated gene expression, laminar similarity, and receptor similarity show more stable community solutions, where larger changes in $\gamma$ are required for the network to split itself into more communities. This suggests that molecular connectivity modes can be described from the perspective of a small number (<10) of modules. We show 1 possible consensus community detection solution for each network in Fig 5E, which demonstrates that the modular organization and gradient decomposition of networks tend to be closely aligned. Collectively, this shows that each connectivity mode has a unique gradient decomposition and community structure.

## Fusing connectivity modes

Each connectivity mode that we have studied so far represents a single scale of organization describing distinct but related interregional relationships. Given that the brain is integrated, how do these connectivity modes layer onto one another to support brain structure and function? To address this questions, we apply an unsupervised learning technique, similarity network fusion (SNF), to merge all 7 connectivity modes into a single multimodal network (Fig 6A) [92]. SNF iteratively fuses each connectivity mode in a manner that strengthens edges

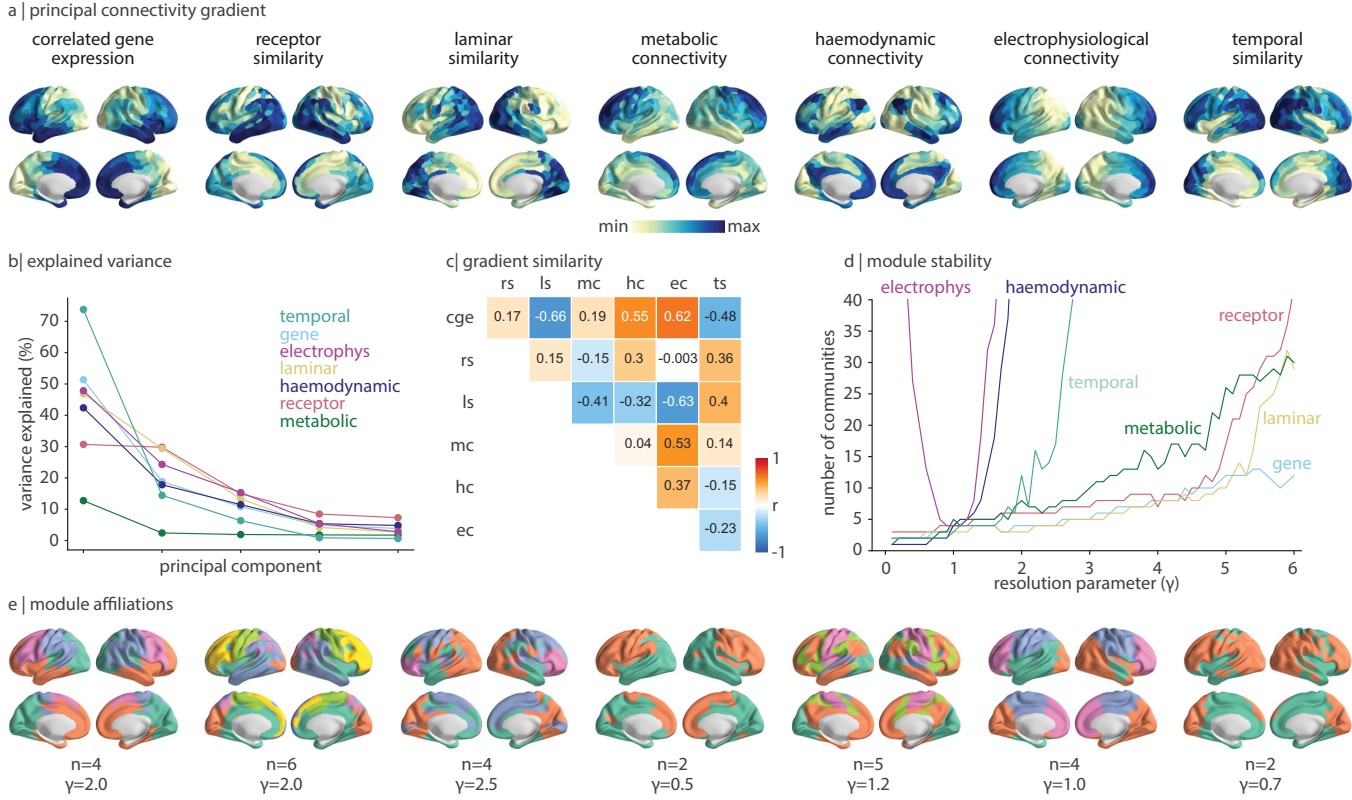

**Fig 5. Gradients and modules of connectivity modes.** (a) The first principal component ("gradient") of each connectivity mode is shown on the cortex. (b) The percent variance explained for the first 5 principal components of each connectivity mode. (c) The Pearson's correlation between every pair of network gradients, visualized as a heatmap. CGE, correlated gene expression; RS, receptor similarity; LS, laminar similarity; MC, metabolic connectivity; HC, haemodynamic connectivity; EC, electrophysiological connectivity; TS, temporal similarity. (d) The Louvain community detection algorithm is applied to each connectivity mode across different resolution parameters ($0.1 \leq \gamma \leq 6.0$, in intervals of 0.1) and the number of ensuing communities is plotted as a function of $\gamma$. (e) For each connectivity mode, we show a single community detection solution for a specified $\gamma$, and we indicate the number of communities ($n$). The data underlying this figure can be found at https://github.com/netneurolab/hansen_many_networks.

that are consistently strong and weakens inconsistent (or consistently weak) edges, while giving each connectivity modality equal influence on the fusion processes. Altogether, the fused network represents a data-driven integration of each level of cortical connectivity.

The fused network's strongest edges exist between regions within somatomotor and visual cortex (Fig 6B, bottom), likely reflecting the conserved molecular and dynamic composition of these phylogenetically older cortical regions. Meanwhile, cortical regions with the greatest weighted degree exist in anterior temporal and superior frontal cortex (Fig 6B, right). The fused network exhibits nonrandom network organization including strong homotopic connectivity and a negative exponential relationships with distance (Fig 6B left, c). In addition, structurally connected edges have significantly stronger edge weight than non-connected edges, against a degree- and edge-length preserving structural null (Fig 6D). Finally, the fused network demonstrates a greater correlation between edge weight and weighted structural connectivity than any of the individual connectivity modes ($r = 0.53$). This shows how combining interregional similarity across multiple scales better reflects anatomical connectivity than any single perspective of interregional similarity [10]. This may be because regions that are similar across multiple scales are more likely to be connected or because brain connectivity gives rise to shared biological features.

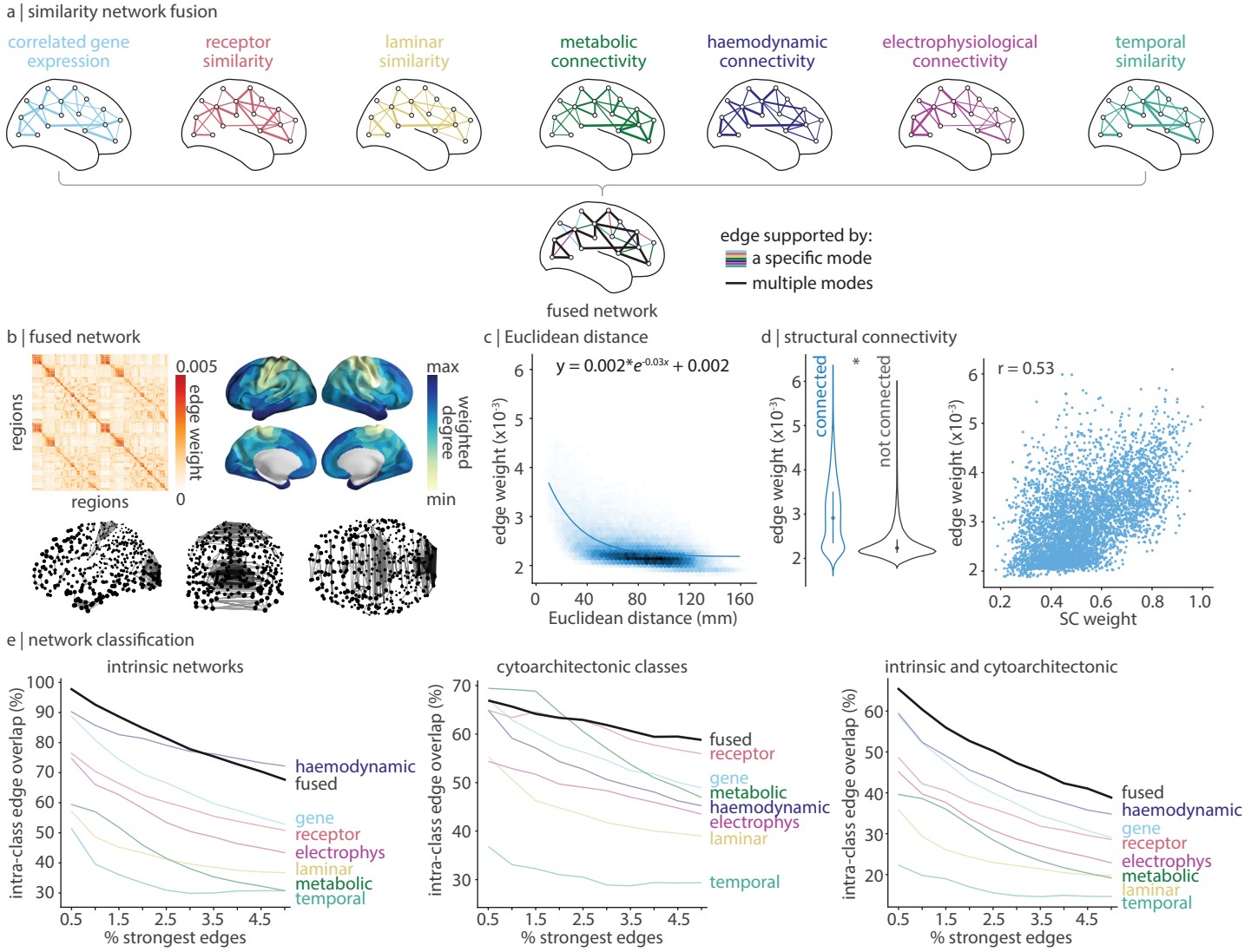

**Fig 6. Network fusion.** SNF was applied to all 7 connectivity modes to construct a single integrated network [92,169]. (a) Toy example of SNF. SNF iteratively combines the 7 connectivity modes in a manner that gives more weight to edges between observations that are consistently high-strength across data types (black edges). (b) The fused network. We show the matrix-representation of the network (left), the top 0.5% strongest edges of the network (bottom), and the weighted degree of each brain region (right). Note that the edge weights in the fused network is a byproduct of the iterative multiplication and normalization steps (see Methods for details) and therefore can become very small. Greater edge magnitude represents greater similarity (no negative edges exist, by design). (c) Edge weight decreases exponentially with Euclidean distance. (d) Structurally connected edges have greater edge weight than edges without an underlying structural connection, against a degree and edge-length preserving null model [56] (left), and is correlated with structural connectivity (right). (e) For a varying threshold of strongest edges (0.5%–5% in 0.5% intervals), we calculate the proportion of edges that connect 2 regions within the same intrinsic network (left), cytoarchitectonic class (middle), and the union of intrinsic networks and cytoarchitectonic classes (right). The data underlying this figure can be found at https://github.com/netneurolab/hansen_many_networks. SNF, similarity network fusion.

We next asked whether the strongest edges in the fused network exist between functionally and cytoarchitectonically similar cortical regions (Fig 6E). We find that nearly all (97.7%) of the top 0.5% strongest edges in the fused network are between regions within the same functional network. In fact, the fused network outperforms haemodynamic connectivity—the connectivity mode for which these intrinsic functional networks are designed and optimized. Likewise, for cytoarchitectonic classes, we find that the fused network retains more intra-class edges than any other network when the number of strongest edges considered is ≥2.5%. Since

the fused network represents an integrated connectivity mode, we asked whether the strongest edges of the fused network might simultaneously maximize intrinsic and cytoarchitectonic intra-class edges. Indeed, when considering the top 0.5% to 5.0% strongest edges, the number of edges that exist between regions in the same intrinsic and cytoarchitectonic classes is consistently greatest for the fused network. Altogether, the fused network maps onto intrinsic networks and cytoarchitectonic classes better than any individual network. This demonstrates how large-scale phenomena emerge from a confluence of multiple microscopic determinants.

## Sensitivity and replication analysis

Finally, to ensure results are not dependent on the parcellation, we repeated all analyses (except Fig 4 which depends on the 68-region Desikan–Killiany parcellation) using the 100-region Schaefer parcellation and the 68-region Desikan–Killiany parcellation [48,93,94]. We find similar results under these alternative parcellations (S7 Fig). These coarser resolutions reveal dense frontal inter-connectivity in the metabolic network, which was not visible at the 400-node parcellation likely due to smoothing effects in dynamic PET data. Furthermore, we share all 7 connectivity modes at these 3 parcellations (Schaefer-400, Schaefer-100, Desikan–Killany-68) in hopes of facilitating integrative connectome analyses in the future (https://github.com/netneurolab/hansen_many_networks).

## Discussion

This work integrates multiple representations of cortical connectivity to establish how diverse connectivity modes contribute to cortical structure and function. We systematically document the common organizational patterns of connectivity modes, as well as their unique contributions to structure and geometry. We find that molecular connectivity modes amplify disease exposure resulting in spatial patterns of abnormal cortical thickness. We show that connectivity modes demonstrate diverse dominant gradients and modular structure. Finally, we derive a multimodal, multiscale network by parsimoniously integrating multiple connectivity modes.

Connectomics—the study of relationships between neural elements across multiple scales— is an important and popular paradigm in neuroscience [3,95,96]. Numerous technological and analytic methods have been developed to reconstruct interregional relationships, some focused on physical wiring, others on molecular similarity, and others still on coherence between regional neural activity. Despite being rooted in common questions, these connectivity modes are often studied in separate literatures. What network features are unique or common to each connectivity mode remains unknown and the practice of studying connectivity modes separately has precluded a truly integrated understanding of interregional relationships.

Detailed comprehensive datasets alongside better data sharing practices have made multimodal, integrative approaches to studying human brain connectivity more feasible [89,97–99]. Examples include comparisons of dynamic FDG-PET and BOLD connectivity [15,29], BOLD connectivity and electrophysiology [28,44,100,101], structural and BOLD connectivity [60,102], and correlated gene expression and structural connectivity [103]. Combining connectivity modes has also been used to better resolve clusters of functional activation in BOLD data [104] and inform the application of deep brain stimulation to psychiatric and neurological diseases [105,106]. Encouragingly, previous work has found that incorporating multiple perspectives of brain connectivity can result in novel discoveries, including improved generative models of brain connectivity [107], structure-function coupling [23,38], epicenters of transdiagnostic alterations [33,34], and the characterization of homophilic wiring principles [108].

Although these integrative approaches open exciting new questions about brain organization, an important challenge remains: How do we ensure that conclusions are rooted in the

underlying biology rather than assumptions and idiosyncrasies of individual data modalities? We attempted to mitigate this challenge by repeating analyses using other analytic choices, applying conservative null models, normalizing each connectivity mode prior to analysis (Fisher's *r*-to-*z* transformation), and rank-transforming edges to facilitate comparison of edge strengths across data types. This provides a level of confidence but is by no means an exhaustive verification that data types do not influence results. Indeed, each dataset is accompanied by its own set of limitations including instances of false positives and negatives in diffusion tractography [109–111], nonspecific binding for some PET tracers [112], and heterogeneous patterns of signal to noise ratios across all imaging types. As open datasets are created and shared, it will become more feasible to determine how results are influenced by processing choices and imaging modalities.

The study of connectomics has been dominated by a focus on structural and haemodynamic connectivity. This has resulted in the assumption that, at the level of the whole-cortex, homologous [113], spatially proximal [55], and structurally connected [54] brain regions tend to be more similar. By systematically integrating 7 multiscale perspectives of cortical connectivity, we can conclude in a more systematic and comprehensive manner that these properties are indeed fundamental to cortical organization but that there is considerable variation across connectivity modes. For example, the negative exponential relationship with distance is almost linear for molecular connectivity modes, especially when we consider geodesic instead of Euclidean distance (S2 Fig). A second assumption is that diverse brain features should all follow the functionally defined unimodal-transmodal hierarchical gradient and can be organized in terms of intrinsic resting-state networks [49,86,114–116]. However, we find that microscale connectivity modes (e.g., correlated gene expression, receptor similarity) are well delineated by a partition based on cytoarchitectonic classes, whereas dynamic connectivity modes (e.g., haemodynamic connectivity, electrophysiological connectivity) fit better into intrinsic functional systems. Indeed, connectivity modes are poorly correlated with one another, suggesting that each connectivity mode provides a fundamentally different but important view of how cortical regions participate in neural circuits at different spatial and temporal scales [117].

In an effort to understand which cortical regions are consistently central across many levels of description, we identify a set of cross-modal hubs. Brain hubs are conventionally defined as regions with a relatively large number of structural connections, but this definition ignores the multiscale character of brain networks. Indeed, we find that hub profiles are not redundant across biological mechanisms. Instead, we identify a subset of cortical regions that are uniquely central across multiple levels of description. These cross-modal hubs exist in the precuneus, supramarginal gyrus, and dorsolateral prefrontal cortex: association regions that expand in surface area and develop into more differentiated eulaminate (6-layer) cortex during evolution [18,73,118]. This suggests that phylogenetic structural modifications—including increased cellular complexity and density [119]—may support integration of information across multiple biological scales, resulting in higher-level cognition including language, planning, and complex executive functions. Interestingly, these regions are distinct from the anatomically central (e.g., limbic) regions that were previously hypothesized to be integrative general-domain hubs, based on multiple measures of centrality calculated on the structural connectome [120]. Future work should investigate more deeply how structural centrality is aligned with biological feature similarity. Altogether, cross-modal hubs open a new perspective on hub function: Instead of being rooted only in high structural connectivity, hubs can be classified according to their participation in different biological systems [65].

Integrative connectomics opens the possibility of benchmarking and comparing biological mechanisms to one another. For example, we consistently identify a dichotomy between molecular (e.g., correlated gene expression, receptor similarity, laminar similarity) and

dynamic (e.g., haemodynamic and electrophysiological connectivity) modes. First, molecular feature similarity is significantly increased for links between regions of the brain's rich club: high-degree regions that show dense inter-connectivity which is thought to improve global communication efficiency and integration [7]. A transcriptional signature of rich club connectivity was previously shown to be driven by genes involved in metabolism, supporting the theory that the brain's rich club is energetically expensive [63,121,122]. Interestingly, we find that metabolic connectivity is increased in rich links, suggesting that the rich club is also synchronized in its energy consumption [66,67].

Second, molecular feature similarity—particularly correlated gene expression and receptor similarity—best explains the spatial patterning of multiple cortical disease abnormalities. Recent work has explored the idea that multiple pathologies spread trans-synaptically, including misfolded proteins, aberrant neurodevelopmental signals, and excitotoxic electrical discharge, resulting in patterns of pathology that reflect the underlying structural architecture of the brain [11,81]. Here, we consider the possibility that shared vulnerability to disease arises not just from structural connectivity but also from multiscale biological attributes [33]. We use changes in cortical thickness as the marker of potential pathology and find that when disease exposure is informed by transcriptional and receptor similarity, we can reproduce the cortical profile of multiple diseases ($r > 0.5$ for most). We also find evidence that molecular dissimilarity may serve as a mitigating factor of pathological spread (S5B Fig), although more work is necessary to determine the link between regional dissimilarity and pathology. The consistent primacy of molecular connectivity modes demonstrates that mapping cortical connectivity from the perspective of underlying microscale features—gene transcription, receptor density, cellular composition—is just as, if not more, informative than oft-studied dynamical modes such as haemodynamic connectivity. This analysis can be extended in future work by studying the connectivity modes of patient populations and will hopefully motivate future causal work linking molecular mechanisms to the spreading of pathological markers in the brain.

Lastly, we examine the gradient and modular organization of connectivity modes. Low-dimensional topographical representations of cortical features, whether spatially continuous (gradients) or discrete (modules), present insight on how different levels of cortical organization are aligned with one another. For example, graded changes in the proportion of neural projections originating from upper versus lower laminar layers has been related to gradients of pyramidal neuron soma size, number of synaptic boutons, number of vesicles, amount of neurotransmitter release, and firing rate, providing a comprehensive explanation for the laminar origin of cortico-cortical connections [123]. Likewise [22], showed that autoradiography-derived gradients of receptor density reflect excitatory/inhibitory and ionotropic/metabotropic ratios and follow a sensory-association functional hierarchy. Here, we view local molecular and dynamic features from the perspective of a network to study the organization of cortical "connectivity," rather than axes of variation of the underlying data [22,23,124]. This approach lets us apply not only gradient decomposition but also community detection—rarely used in brain imaging outside of structural and haemodynamic connectivity—to the networks. We find that connectivity modes have unique gradient and modular decomposition which means it is not sufficient to assume a single spatial organization for the cortex. Interestingly, previous work has found that fMRI-derived functional communities can themselves be diverse: they fluctuate over time, during tasks, and throughout hormonal cycles [125–127]. These temporary changes in network organization may reflect the diverse modular organization of underlying molecular mechanisms. The biological origins of the diversity in spatial gradients will be an important direction for future research [88,128].

Throughout this report, we have illustrated how cortical connectivity can be extended from neural wiring to many complementary perspectives of interregional relationships. We focus

on densely sampled data across the whole-cortex to make general claims about patterns of cortical organization. Most of the data employed are derived from in vivo neuroimaging in relatively large samples of healthy adult humans. However, these questions about multiscale cortical organization can—and have, for decades—be asked with more anatomical specificity using methods such as cell staining and tract-tracing in small samples of ex vivo brains, generally from model organisms [17,18,21,129]. Such studies have demonstrated that neurons make projection patterns that are tightly linked to the cellular architecture of the cortex, including the laminar differentiation of the source and target of a neural projection [19] (e.g., the Structural Model, reviewed in [18]). Intertwined with laminar differentiation and tract-tracing projection patterns is also the phylogenetic age of the cortical region [119], markers of plasticity and stability [24], and likely also receptor architecture [130,131]. Furthermore, the field of developmental biology presents fundamental organizational principles for how the brain develops its spatial organization and topology [132]. Large-scale neuroimaging connectomic studies complement biological and neuroanatomical studies by extending predictions to the scale of the whole human cortex and across many more brain phenotypes. For example, we confirm that laminar similarity is related to connectivity and the brain's rich club [57], but extend this to gene expression, receptor architecture, and metabolism. The synergy between neuroanatomical and imaging fields is necessary to fully capture interregional relationships across multiple layers of description.

The present work should be considered alongside some methodological considerations. First, the results are only representative of the 7 included connectivity modes; future work should replicate the findings in similar connectivity modes derived from external datasets, as well as extend this work into additional forms of connectivity. One exciting avenue would be to annotate structural connectomes with measures of myelin or axon caliber derived from quantitative MRI such as magnetization transfer (MT), T1 relaxation rate (R1), or axon diameter [133–135]. Second, each connectivity matrix is dependent on the quality of the imaging modality, and each imaging method operates at a unique spatial and temporal resolution. Results may therefore be influenced by differences in how the data are acquired. In addition to this, the group-consensus structural network that was used throughout the analyses (in particular Figs 1, 2, 4 and 6) was reconstructed from diffusion spectrum imaging and tractography, which is prone to false-positives and false-negatives [109,111]. We tried to mitigate this by running extensive sensitivity analyses. Third, in an effort to make correlated gene expression comparable to the other modes, data interpolation and mirroring was conducted, potentially biasing this network towards homotopic connections. Fourth, connectivity modes are compiled across different individuals of varying ages, sex ratios, and handedness. Results are therefore limited to group-averages and motivate future deep phenotyping studies of the brain across multiple scales and modalities. Fifth, the chosen functionally defined Schaefer parcellation used for all main analyses may better reflect functional networks (e.g., haemodynamic connectivity, electrophysiological connectivity, temporal similarity) than molecular networks. We aimed to mitigate this limitation by repeating analyses using an anatomically defined parcellation (Desikan–Killiany; S7 Fig). Future integrative parcellations designed using multiple brain phenotypes would be ideal for studying multiscale, multimodal connectivity modes.

Altogether, this work combines 7 perspectives of cortical connectivity from diverse spatial scales and imaging modalities including gene expression, receptor density, cellular composition, metabolic consumption, haemodynamic activity, electrophysiology, and time series features. We demonstrate both the similar and complementary ways in which connectivity modes reflect cortical geometry, structure, and disease. This serves as a step towards the next-generation integrative, multimodal study of cortical connectivity.

## Methods

### Connectivity modes

We construct cortical connectivity modes for 7 different brain features: gene expression, receptor density, lamination, glucose uptake, haemodynamic activity, electrophysiological activity, and temporal profiles. Each connectivity mode is defined across 400 cortical regions, ordered according to 7 intrinsic networks (visual, somatomotor, dorsal attention, ventral attention, limbic, frontoparietal, and default mode), separated by hemispheres (left, right) [48]. This functionally defined parcellation scheme was chosen because the parcels are approximately equal in size and parcel boundaries respect both functional boundaries (as determined by resting-state and task-based fMRI) as well as histological boundaries [48]. Nonetheless, we repeated the analyses using the coarser 100-region Schaefer parcellation as well as an anatomically defined 68-region Desikan–Killiany parcellation and found consistent results (S7 Fig; parcellated connectivity modes all available at https://github.com/netneurolab/hansen_many_networks). To facilitate comparison between connectivity modes, each connectivity mode is normalized using Fisher's *r*-to-*z* transform ($z = \text{arctanh}(r)$). We describe the construction of each connectivity mode in detail below.

**Correlated gene expression.** Correlated gene expression represents the transcriptional similarity between pairs of cortical regions. Regional microarray expression data were obtained from 6 postmortem brains (1 female, ages 24.0–57.0, 42.50±13.38) provided by the AHBA (https://human.brain-map.org [40]). Data were processed with the abagen toolbox (version 0.1.1; https://github.com/rmarkello/abagen [136]) using a 400-region volumetric atlas in MNI space.

First, microarray probes were reannotated using data provided by [137]; probes not matched to a valid Entrez ID were discarded. Next, probes were filtered based on their expression intensity relative to background noise [138], such that probes with intensity less than the background in $\geq$50% of samples across donors were discarded, yielding 31,569 probes. When multiple probes indexed the expression of the same gene, we selected and used the probe with the most consistent pattern of regional variation across donors (i.e., differential stability [139]), calculated with:

$$\Delta_S(p) = \frac{1}{\binom{N}{2}} \sum_{i=1}^{N-1} \sum_{j=i+1}^{N} r\left[B_i(p), B_j(p)\right],$$

where $\rho$ is Spearman's rank correlation of the expression of a single probe, $p$, across regions in 2 donors $B_i$ and $B_j$, and $N$ is the total number of donors. Here, regions correspond to the structural designations provided in the ontology from the AHBA.

The MNI coordinates of tissue samples were updated to those generated via nonlinear registration using the Advanced Normalization Tools (ANTs; https://github.com/chrisfilo/alleninf). To increase spatial coverage, tissue samples were mirrored bilaterally across the left and right hemispheres [103]. Samples were assigned to brain regions in the provided atlas if their MNI coordinates were within 2 mm of a given parcel. If a brain region was not assigned a tissue sample based on the above procedure, every voxel in the region was mapped to the nearest tissue sample from the donor in order to generate a dense, interpolated expression map. The average of these expression values was taken across all voxels in the region, weighted by the distance between each voxel and the sample mapped to it, in order to obtain an estimate of the parcellated expression values for the missing region. All tissue samples not assigned to a brain region in the provided atlas were discarded.

Inter-subject variation was addressed by normalizing tissue sample expression values across genes using a robust sigmoid function [46]:

$$x_{\text{norm}} = \frac{1}{1 + \exp\left(-\frac{(x_g - \langle x_g \rangle)}{\text{IQR}_x}\right)},$$

where $\langle x \rangle$ is the median and $\text{IQR}_x$ is the normalized interquartile range of the expression of a single tissue sample across genes. Normalized expression values were then rescaled to the unit interval:

$$x_{\text{scaled}} = \frac{x_{\text{norm}} - \min(x_{\text{norm}})}{\max(x_{\text{norm}}) - \min(x_{\text{norm}})}.$$

Gene expression values were then normalized across tissue samples using an identical procedure. Samples assigned to the same brain region were averaged separately for each donor, yielding a regional expression matrix for each donor with 400 rows, corresponding to brain regions, and 15,633 columns, corresponding to the retained genes. A threshold of 0.1 was imposed on the differential stability of each gene, such that only stable genes were retained for future analysis, resulting in 8,687 retained genes.

Finally, the region × region correlated gene expression matrix was constructed by correlating (Pearson's $r$) the normalized gene expression profile at every pair of brain regions. This matrix was then normalized using Fisher's $r$-to-$z$ transform.

**Receptor similarity.** Receptor similarity indexes the degree to which the receptor density profiles at 2 cortical regions are correlated. Conceptually, it can be thought of as how similarly 2 cortical regions might "hear" the same neural signal. PET tracer images for 18 neurotransmitter receptors and transporters were obtained from [23] and neuromaps (v0.0.1, https://github.com/netneurolab/neuromaps [89]). The receptors/transporters span 9 neurotransmitter systems including: dopamine (D1, D2, DAT), norepinephrine (NET), serotonin (5-HT1A, 5-HT1B, 5-HT2, 5-HT4, 5-HT6, 5-HTT), acetylcholine ($\alpha_4\beta_2$, M1, VAChT), glutamate (mGluR5), GABA (GABAA), histamine (H3), cannabinoid (CB1), and opioid (MOR). Tracer names and number of participants (with number of females in parentheses) are listed for each receptor in S1 Table. Each PET tracer image was parcellated to 400 cortical regions and z-scored. A region-by-region receptor similarity matrix was constructed by correlating (Pearson's $r$) receptor profiles at every pair of cortical regions. This matrix was then normalized using Fisher's $r$-to-$z$ transform.

**Laminar similarity.** Laminar similarity is estimated from histological data and aims to uncover how similar pairs of cortical regions are in terms of cellular distributions across the cortical laminae. Specifically, we use data from the BigBrain, a high-resolution (20 $\mu$m) histological reconstruction of a postmortem brain from a 65-year-old male [14,41]. Cell-staining intensity profiles were sampled across 50 equivolumetric surfaces from the pial surface to the white mater surface to estimate laminar variation in neuronal density and soma size. Intensity profiles at various cortical depths can be used to approximately identify boundaries of cortical layers that separate supragranular (cortical layers I to III), granular (cortical layer IV), and infragranular (cortical layers V to VI) layers.

The data were obtained on *fsaverage* surface (164k vertices) from the BigBrainWarp toolbox [140] and were parcellated into 400 cortical regions according to the Schaefer-400 atlas [48]. The region × region laminar similarity matrix was calculated as the partial correlation (Pearson's $r$) of cell intensities between pairs of cortical regions, after correcting for the mean intensity across cortical regions. Laminar similarity was first introduced in [14] and has also

been referred to as "microstructure profile covariance." This matrix was then normalized using Fisher's $r$-to-$z$ transform.

**Metabolic connectivity.** Metabolic connectivity indexes how similarly 2 cortical regions metabolize glucose over time and therefore how similarly 2 cortical regions consume energy. Volumetric 4D PET images of $[F^{18}]$-fluordoxyglucose (FDG, a glucose analogue) tracer uptake over time were obtained from [42]. Specifically, 26 healthy participants (77% female, 18 to 23 years old) were recruited from the general population and underwent a 95-min simultaneous MR-PET scan in a Siemens (Erlangen) Biograph 3-Tesla molecular MR scanner. Participants were positioned supine in the scanner bore with their head in a 16-channel radiofrequency head coil and were instructed to lie as still as possible with eyes open and think of nothing in particular. FDG (average dose 233 MBq) was infused over the course of the scan at a rate of 36 mL/h using a BodyGuard 323 MR-compatible infusion pump (Caesarea Medical Electronics, Caesarea, Israel). Infusion onset was locked to the onset of the PET scan. This data has been validated and analyzed previously in [15,29].

PET images were reconstructed and preprocessed according to [15]. Specifically, the 5,700-second PET time series for each subject was binned into 356 3D sinogram frames each of 16-s intervals. The attenuation for all required data was corrected via the pseudo-CT method [141]. Ordinary Poisson-Ordered Subset Expectation Maximization algorithm (3 iterations, 21 subsets) with point spread function correction was used to reconstruct 3D volumes from the sinogram frames. The reconstructed DICOM slices were converted to NIFTI format with size $344 \times 344 \times 127$ (voxel size: $2.09 \times 2.09 \times 2.03$ mm$^3$) for each volume. A 5 mm FWHM Gaussian postfilter was applied to each 3D volume. All 3D volumes were temporally concatenated to form a 4D ($344 \times 344 \times 127 \times 356$) NIFTI volume. A guided motion correction method using simultaneously acquired MRI was applied to correct the motion during the PET scan; 225 16-s volumes were retained commencing for further analyses.

Next, the 225 PET volumes were motion corrected (FSL MCFLIRT [142]) and the mean PET image was brain extracted and used to mask the 4D data. The fPET data were further processed using a spatiotemporal gradient filter to remove the accumulating effect of the radiotracer and other low-frequency components of the signal [42]. Finally, each time point of the PET volumetric time series were registered to MNI152 template space using Advanced Normalization Tools in Python (ANTSpy, https://github.com/ANTsX/ANTsPy), parcellated to 400 regions according to the Schaefer atlas, and time series at pairs of cortical regions were correlated (Pearson's $r$) to construct a metabolic connectivity matrix for each subject. A group-averaged metabolic connectome was obtained by averaging connectivity across subjects, and lastly, the matrix was normalized using Fisher's $r$-to-$z$ transform.

**Haemodynamic connectivity.** Haemodynamic connectivity, commonly simply referred to as "functional connectivity," captures how similarly pairs of cortical regions exhibit fMRI BOLD activity at rest [143]. The fMRI BOLD time series picks up on magnetic differences between oxygenated and deoxygenated hemoglobin to measure the haemodynamic response: the oversupply of oxygen to active brain regions [144]. fMRI data were obtained for 326 unrelated participants (age range 22 to 35 years, 145 males) from the HCP (S900 release [43]). All 4 resting state fMRI scans (2 scans (R/L and L/R phase encoding directions) on day 1 and 2 scans (R/L and L/R phase encoding directions) on day 2, each about 15-min long; TR = 720 ms) were available for all participants. fMRI data were preprocessed using HCP minimal preprocessing pipelines [43,145]. Specifically, all 3T fMRI time series were corrected for gradient nonlinearity, head motion using a rigid body transformation, and geometric distortions using scan pairs with opposite phase encoding directions (R/L, L/R) [146]. Further preprocessing steps include co-registration of the corrected images to the T1w structural MR images, brain extraction, normalization of whole brain intensity, high-pass filtering (>2,000 s FWHM; to

correct for scanner drifts), and removing additional noise using the ICA-FIX process [146,147]. The preprocessed time series were then parcellated to 400 cortical brain regions according to the Schaefer atlas [48]. The parcellated time series were used to construct functional connectivity matrices as a Pearson correlation coefficient between pairs of regional time series for each of the 4 scans of each participant. A group-average functional connectivity matrix was constructed as the mean functional connectivity across all individuals and scans. This matrix was then normalized using Fisher's $r$-to-$z$ transform.

**Electrophysiological connectivity.** Electrophysiological connectivity was measured using MEG recordings, which tracks the magnetic field produced by neural currents. Resting state MEG data was acquired for $n$ = 33 unrelated healthy young adults (age range 22 to 35 years) from the HCP (S900 release [43]). The data includes resting state scans of approximately 6-min long and noise recording for all participants. MEG anatomical data and 3T structural MRI of all participants were also obtained for MEG preprocessing.

The present MEG data was first processed and used by [44]. Resting state MEG data was preprocessed using the open-source software, Brainstorm (https://neuroimage.usc.edu/brainstorm/ [148]), following the online tutorial for the HCP dataset (https://neuroimage.usc.edu/brainstorm/Tutorials/HCP-MEG). MEG recordings were registered to individual structural MRI images before applying the following preprocessing steps. First, notch filters were applied at 60, 120, 180, 240, and 300 Hz, followed by a high-pass filter at 0.3 Hz to remove slow-wave and DC-offset artifacts. Next, bad channels from artifacts (including heartbeats, eye blinks, saccades, muscle movements, and noisy segments) were removed using Signal-Space Projections (SSP).

Preprocessed sensor-level data was used to construct a source estimation on HCP's fsLR4k cortex surface for each participant. Head models were computed using overlapping spheres and data and noise covariance matrices were estimated from resting state MEG and noise recordings. Linearly constrained minimum variance (LCMV) beamformers was used to obtain the source activity for each participant. Data covariance regularization was performed and the estimated source variance was normalized by the noise covariance matrix to reduce the effect of variable source depth. All eigenvalues smaller than the median eigenvalue of the data covariance matrix were replaced by the median. This helps avoid instability of data covariance inversion caused by the smallest eigenvalues and regularizes the data covariance matrix. Source orientations were constrained to be normal to the cortical surface at each of the 8,000 vertex locations on the cortical surface, then parcellated according to the Schaefer-400 atlas [48].

After preprocessing and parcellating the data, amplitude envelope correlations were performed between time series at each pair of brain regions, for 6 canonical frequency bands separately (delta (2 to 4 Hz), theta (5 to 7 Hz), alpha (8 to 12 Hz), beta (15 to 29 Hz), low gamma (30 to 59 Hz), and high gamma (60 to 90 Hz)). Amplitude envelope correlation is applied instead of directly correlating the time series because of the high sampling rate (2,034.5 Hz) of the MEG recordings. An orthogonalization process was applied to correct for the spatial leakage effect by removing all shared zero-lag signals [149]. The composite electrophysiological connectivity matrix is the first principal component of all 6 connectivity matrices (vectorized upper triangle) and closely resembles alpha connectivity (S8 Fig). Finally, the matrix underwent Fisher's $r$-to-$z$ transform.

**Temporal profile similarity.** Temporal profile similarity was first introduced by, and obtained from [45], and represents how much 2 cortical regions exhibit similar temporal features, as calculated on fMRI time series. Note that although this connectivity mode is derived from the same imaging modality as haemodynamic connectivity, it is fundamentally different from haemodynamic connectivity as it represents a comprehensive account of dynamic similarity (Pearson's $r$ = 0.24, S3 Fig). This is in contrast to haemodynamic connectivity that

measures the Pearson's correlation between the time series themselves. Specifically, we used the highly comparative time series analysis toolbox, *hctsa* [46,47] to perform a massive feature extraction of the parcellated fMRI time series (see Haemodynamic connectivity) at each brain region of each participant. The *hctsa* package extracted over 7,000 local time series features using a wide range of operations based on time series analysis. The extracted features include, but are not limited to, distributional features, entropy and variability, autocorrelation, time-delay embeddings, and nonlinear features of a given time series. Following the feature extraction procedure, the outputs of the operations that produced errors were removed and the remaining features (6,441 features) were normalized across nodes using an outlier-robust sigmoidal transform. We used Pearson's correlation coefficients to measure the pairwise similarity between the time series features of all possible combinations of cortical areas. As a result, a temporal profile similarity network was constructed for each individual and each run, representing the strength of the similarity of the local temporal fingerprints of cortical areas. This matrix was then normalized using Fisher's *r*-to-*z* transform.

## Structural connectivity

Diffusion weighted imaging (DWI) data were obtained for 326 unrelated participants (age range 22 to 35 years, 145 males) from the HCP (S900 release [43]) [146]. DWI data was preprocessed using the MRtrix3 package [150] (https://www.mrtrix.org/). More specifically, fiber orientation distributions were generated using the multi-shell multi-tissue constrained spherical deconvolution algorithm from MRtrix [151,152]. White matter edges were then reconstructed using probabilistic streamline tractography based on the generated fiber orientation distributions [153]. The tract weights were then optimized by estimating an appropriate cross-section multiplier for each streamline following the procedure proposed by [154] and a connectivity matrix was built for each participant using the 400-region Schaefer parcellation [48]. A group-consensus binary network was constructed using a method that preserves the density and edge-length distributions of the individual connectomes [155–157]. Edges in the group-consensus network were assigned weights by averaging the log-transformed streamline count of nonzero edges across participants. Edge weights were then scaled to values between 0 and 1.

## Disease exposure

Patterns of cortical thickness from the ENIGMA consortium and the *enigma* toolbox were available for 13 neurological, neurodevelopmental, and psychiatric disorders (https://github.com/MICA-MNI/ENIGMA [33,79,80]), including: 22q11.2 deletion syndrome (*N* = 474 participants, *N* = 315 controls) [158], attention-deficit/hyperactivity disorder (ADHD; *N* = 733 participants, *N* = 539 controls) [159], autism spectrum disorder (ASD; *N* = 1,571 participants, *N* = 1,651 controls) [160], idiopathic generalized (*N* = 367 participants), right temporal lobe (*N* = 339 participants), and left temporal lobe (*N* = 415 participants) epilepsies (*N* = 1,727 controls) [161], depression (*N* = 2,148 participants, *N* = 7,957 controls) [162], obsessive-compulsive disorder (OCD; *N* = 1,905 participants, *N* = 1,760 controls) [163], schizophrenia (*N* = 4,474 participants, *N* = 5,098 controls) [164], bipolar disorder (*N* = 1,837 participants, *N* = 2,582 controls) [165], obesity (*N* = 1,223 participants, *N* = 2,917 controls) [166], schizotypy (*N* = 3,004 participants) [167], and Parkinson's disease (*N* = 2,367 participants, *N* = 1,183 controls) [168]. The ENIGMA consortium is a data-sharing initiative that relies on standardized processing and analysis pipelines, such that disorder maps are comparable [79]. Altogether, over 21,000 participants were scanned across the 13 disorders, against almost 26,000 controls. The analysis was limited to adults in all cases except ASD where the abnormal cortical thickness map is only available aggregated across all ages (2 to 64 years). The values for each map

are z-scored effect sizes (Cohen's $d$) of cortical thickness in patient populations versus healthy controls. Imaging and processing protocols can be found at http://enigma.ini.usc.edu/protocols/. Local review boards and ethics committees approved each individual study separately, and written informed consent was provided according to local requirements.

We calculate disease exposure for every disease and network, after masking the network such that all edges with negative strength are assigned a strength of 0. For a given network and disease, disease exposure of a node $i$ is defined as,

$$D_i = \frac{1}{N_i} \sum_{j \neq i, j=1}^{N_i} d_j \times c_{ij},$$

where $N_i$ is the number of positive connections made by region $i$, $d_j$ is the abnormal cortical thickness at region $j$, and $c_{ij}$ is the edge strength between regions $i$ and $j$. This analysis was repeated after regressing the exponential fit in Fig 1B from each network, to ensure results are not driven by distance (S9 Fig).

### Community detection

For each connectivity mode, communities were identified using the Louvain algorithm, which maximizes positive edge strength within communities and negative edge strength between communities [90]. Specifically, brain regions were assigned to communities in a manner that maximizes the quality function

$$Q(\gamma) = \frac{1}{m^+} \left[ w_{ij}^+ - \gamma p_{ij}^+ \right] \delta\left(\sigma_i, \sigma_j\right) - \frac{1}{m^+ + m^-} \sum_{ij} [w_{ij}^+ - \delta p_{ij}^-] \delta\left(\sigma_i, \sigma_j\right),$$

where $w_{ij}^+$ is the network with only positive correlations and likewise for $w_{ij}^-$ and negative correlations. The term $p_{ij}^\pm = (s_i^\pm s_j^\pm)/(2m^\pm)$ represents the null model: the expected density of connections between nodes $i$ and $j$, where $s_i^\pm = \sum_j w_{ij}^\pm$ and $m^\pm = \sum_{i,j>i} w_{ij}^\pm$. The variable $\sigma_i$ is the community assignment of node $i$ and $\delta(\sigma_i, \sigma_j)$ is the Kronecker function and is equal to 1 when $\sigma_i = \sigma_j$ and 0 otherwise. The resolution parameter, $\gamma$, scales the relative importance of the null model, making it easier ($\gamma>1$) or harder ($\gamma<1$) for the algorithm to uncover many communities. In other words, as $\gamma$ increases, increasingly fine network partitions are identified. We tested 60 values of $\gamma$, from $\gamma = 0.1$ to $\gamma = 6.0$, in increments of 0.1. At each $\gamma$, we repeated the algorithm 250 times and constructed a consensus partition, following the procedure recommended in [91].

### Similarity network fusion

First introduced by [92], SNF is a method for combining multiple measurement types for the same observations (e.g., patients, or in our case, brain regions) into a single similarity network where edges between observations represent their cross-modal similarity. For each data source, SNF constructs an independent similarity network, defines the $K$ nearest neighbors for each observation, and then iteratively combines the networks in a manner that gives more weight to edges between observations that are consistently high-strength across data types. We used snfpy (https://github.com/rmarkello/snfpy [169]), an open-source Python implementation of the original SNF code provided by [92]. A brief description of the main steps in SNF follows, adapted from its original presentation in [92].

In the present report, the 7 data sources to be fused are the 7 connectivity modes (correlated gene expression, receptor similarity, laminar similarity, metabolic connectivity, haemodynamic connectivity, electrophysiological connectivity, and temporal similarity). First,

similarity networks for each connectivity mode are constructed where edges are determined using a scaled exponential similarity kernel:

$$\mathbf{W}(i,j) = e^{-\frac{\rho^2(x_i,x_j)}{\mu\epsilon_{i,j}}},$$

where $\mathbf{W}(i,j)$ is the edge weight between regions $i$ and $j$, $\rho(x_i, x_j)$ is the Euclidean distance between regions $i$ and $j$, $\mu \in \mathbb{R}$ is a hyperparameter that is set empirically, and

$$\epsilon_{i,j} = \frac{\bar{\rho}(x_i, N_i) + \bar{\rho}(x_j, N_j) + \rho(x_i, x_j)}{3},$$

where $\bar{\rho}(x_i, N_i)$ is the average distance between $x_i$ and all other regions in the network. Note that $\mu$ is a scaling factor that determines the weighting of edges between regions in the similarity network and is set to $\mu = 0.5$ in the present report.

Next, each $\mathbf{W}$ is normalized such that:

$$\mathbf{P}(i,j) = \begin{cases} \frac{W(i,j)}{2\sum_{k \neq i} W(i,k)}, & j \neq i \\ \frac{1}{2}, & j = i \end{cases}.$$

Finally, a sparse matrix $\mathbf{S}$ of the $K$ nearest (i.e., strongest) neighbors is constructed:

$$\mathbf{S}(i,j) = \begin{cases} \frac{W(i,j)}{\sum_{k \in N_i} W(i,k)}, & j \in N_i \\ 0, & \text{otherwise} \end{cases}.$$

In other words, the matrix $\mathbf{P}$ encodes the full information about the similarity of each region to all other regions (within a given connectivity mode), whereas $\mathbf{S}$ encodes only the similarity of the $K$ most similar regions to each region. $K$ is SNF's second hyperparameter, which we set to one tenth the number of regions in the network [40].

The similarity networks are then iteratively fused. At each iteration, the matrices are made more similar to each other via:

$$\mathbf{P}^{(v)} = \mathbf{S}^{(v)} \times \frac{\sum_{k \neq v} \mathbf{P}^{(k)}}{m-1} \times (\mathbf{S}^{(v)})^{\mathrm{T}}, v = \{1, 2, \ldots m\}.$$

After each iteration, the generated matrices are renormalized as in the normalization step. Fusion stops when the matrices have converged or after a specified number of iterations (in our case, 20). Regions $x_i$ and $x_j$ will likely be neighbors in the fused network if they are neighbors in multiple similarity networks. Furthermore, if $x_i$ and $x_j$ are not very similar in one data type, their similarity can be expressed in another data type. Note that the edge weights in the final fused network are a byproduct of the iterative multiplication and normalization steps, and therefore can become very small. Greater edge magnitude represents greater similarity (no negative edges exist, by design).

After the fusion process, we confirm that no single network exerts undue influence on the final fused network by repeating the fusion process while excluding a single network. The minimum correlation (Spearman $r$) between the leave-one-out fused network and the complete fused network is 0.958. In addition to this, we confirm that alternative $K$ and $\mu$ parameters would not make large difference to the fused network. We test $K \in [20, 59]$ and $\mu \in [0.3, 0.8]$ and find that these alternative fused networks are highly correlated with the original (minimum Spearman $r = 0.924$).

## Null models

**Spin tests.**    Spatial autocorrelation-preserving permutation tests were used to assess statistical significance of associations across cortical regions, termed "spin tests" [170–172]. We created a surface-based representation of the parcellation on the FreeSurfer fsaverage surface, via files from the Connectome Mapper toolkit (https://github.com/LTS5/cmp). We used the spherical projection of the fsaverage surface to define spatial coordinates for each parcel by selecting the coordinates of the vertex closest to the center of the mass of each parcel [36]. These parcel coordinates were then randomly rotated, and original parcels were reassigned the value of the closest rotated parcel (10,000 repetitions). Parcels for which the medial wall was closest were assigned the value of the next most proximal parcel instead. The procedure was performed at the parcel resolution rather than the vertex resolution to avoid upsampling the data and to each hemisphere separately.

**Network randomization.**    Structural networks were randomized using a procedure that preserves the density, edge length, degree distributions of the empirical network [56,172]. Edges were binned according to Euclidean distance (10 bins). Within each bin, pairs of edges were selected at random and swapped, for a total number of swaps equal to the number of regions in the network multiplied by 20. This procedure was repeated 1,000 times to generate 1,000 null structural networks, which were then used to generate null distributions of network-level measures.

## Supporting information

**S1 Fig. Distribution of normalized edge weight for each connectivity mode.**
(PDF)

**S2 Fig. Relationship between edge strength and geodesic distance.**
(PDF)

**S3 Fig. Edge-wise correspondence between connectivity modes.**
(PDF)

**S4 Fig. Alternative views of the strongest edges in each connectivity mode.**
(PDF)

**S5 Fig. Investigating the influence of negative edges in disease exposure.**
(PDF)

**S6 Fig. Contributions of connectivity modes including the fused network to disease vulnerability.**
(PDF)

**S7 Fig. Replication using alternative parcellation.**
(PDF)

**S8 Fig. MEG connectomes across frequency bands.**
(PDF)

**S9 Fig. Contributions of connectivity modes to disease vulnerability.**
(PDF)

**S1 Table. Neurotransmitter receptors and transporters included in receptor similarity.**
(PDF)

## Acknowledgments

We thank Vincent Bazinet, Zhen-Qi Liu, Filip Milisav, Laura Suarez, and Andrea Luppi for their comments and suggestions on the manuscript; the Monash University Neural Systems and Behviour Lab for insightful discussion; and all individuals involved in making the employed open-source datasets available.

## Author Contributions

**Conceptualization:** Justine Y. Hansen, Bratislav Misic.

**Data curation:** Justine Y. Hansen, Golia Shafiei, Katharina Voigt, Emma X. Liang, Sylvia M. L. Cox, Marco Leyton, Sharna D. Jamadar.

**Formal analysis:** Justine Y. Hansen.

**Funding acquisition:** Justine Y. Hansen, Bratislav Misic.

**Investigation:** Justine Y. Hansen.

**Project administration:** Bratislav Misic.

**Supervision:** Bratislav Misic.

**Visualization:** Justine Y. Hansen.

**Writing – original draft:** Justine Y. Hansen, Bratislav Misic.

**Writing – review & editing:** Justine Y. Hansen, Golia Shafiei, Katharina Voigt, Emma X. Liang, Sylvia M. L. Cox, Marco Leyton, Sharna D. Jamadar, Bratislav Misic.

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
