## [Editor Report · Decision Letter 0]

31 Mar 2023

Dear Dr Hansen, 

Thank you for submitting your manuscript entitled "Integrating multimodal, multiscale connectivity blueprints of the cerebral cortex" for consideration as a Research Article by PLOS Biology.

Your manuscript has now been evaluated by the PLOS Biology editorial staff as well as by an academic editor with relevant expertise and I am writing to let you know that we would like to send your submission out for external peer review. I should note that, while we are interested in the study, we have yet to make a firm call about whether the insights offered here go sufficiently beyond previous work in this area to provide the strength of conceptual advance that we aim to publish at PLOS Biology. So we will be looking for enthusiasm from the reviewers regarding that point. 

Before we can send your manuscript to reviewers, we need you to complete your submission by providing the metadata that is required for full assessment. To this end, please login to Editorial Manager where you will find the paper in the 'Submissions Needing Revisions' folder on your homepage. Please click 'Revise Submission' from the Action Links and complete all additional questions in the submission questionnaire.

Once your full submission is complete, your paper will undergo a series of checks in preparation for peer review. After your manuscript has passed the checks it will be sent out for review. To provide the metadata for your submission, please Login to Editorial Manager (https://www.editorialmanager.com/pbiology) within two working days, i.e. by Apr 04 2023 11:59PM.

Kind regards,

Luke

Lucas Smith, Ph.D.

Associate Editor

PLOS Biology

lsmith@plos.org

---

## [Decision Letter · Decision Letter 1]

22 May 2023

Dear Dr Hansen,

Thank you for your patience while your manuscript "Integrating multimodal, multiscale connectivity blueprints of the cerebral cortex" was peer-reviewed at PLOS Biology. It has now been evaluated by the PLOS Biology editors, an Academic Editor with relevant expertise, and by several independent reviewers. In light of the reviews, which you will find at the end of this email, we would like to invite you to revise the work to thoroughly address the reviewers' reports.

As you will see below, while the reviewers appreciate the importance of the topic, and reviewers 2 and 3 comment that the study provides interesting insights, they have raised a number of important concerns which need to be addressed before we can consider your study further.

Reviewer 1 has highlighted that the claims of novelty appear to be overstated and we think his concerns call into question whether the study offers a sufficient advance for PLOS Biology. We therefore think that, in order for the study to be suitable for further consideration at the journal, your manuscript would need to be critically reviewed and revised in the light of the large body of classical and recent studies on principles of ‘connectivity modes’ in humans and animal models. In a new version as well as in your response to reviewers, we think you should state explicitly what new key insights and advances the study has yielded when compared to prior work, (for instance on the Structural Model of Connections by Barbas, and also when honestly compared with your own previous studies using similar data and approaches). We do not think it is enough that the present study is more comprehensive than previous ones - instead, we think it would be necessary to demonstrate that new principles of cortical organization are revealed here. This demonstration may, in fact, not just require additional review of the literature and additional discussion, but likely also additional analyses.

Given the extent of revision needed, we cannot make a decision about publication until we have seen the revised manuscript and your response to the reviewers' comments. Your revised manuscript is likely to be sent for further evaluation by all or a subset of the reviewers.

**IMPORTANT - SUBMITTING YOUR REVISION**

*Re-submission Checklist*

*Published Peer Review*

*PLOS Data Policy*

*Blot and Gel Data Policy*

Sincerely,

Luke

Lucas Smith, Ph.D.

Associate Editor

PLOS Biology

lsmith@plos.org

REVIEWS:

Reviewer #1, Miguel Angel Garcia-Cabezas (note, Reviewer 1 has signed this review): 

The manuscript entitled "Integrating multimodal, multiscale connectivity blueprints of the cerebral cortex" analyze seven datasets of human cerebral cortex structure and function to, in the words of the authors, "assemble a comprehensive multiscale wiring blueprint of the cerebral cortex". The datasets analyzed comprise gene expression data (Allen Human Brain Atlas), receptor density data (obtained from PET-Scan studies), laminar architecture (BigBrain Atlas), metabolism (glucose uptake estimated by FDG-PET), electrophysiology (Resting state MEG from the Human Connectome Project), and temporal fingerprints (hemodynamic resting state measured by fMRI BOLD form the Human Connectome Project and temporal profile similarity). The authors called these datasets "connectivity modes"; thus, gene expression, receptor density, and laminar architecture are "molecular" or "microscale" connectivity modes whereas hemodynamic connectivity, metabolism, and electrophysiology are "dynamic" connectivity modes. 

The major finding of this manuscript is that molecular connectivity modes are well aligned with rich club hubs and selective vulnerability to neuropsychiatric disorders. 

I am a classic neuroanatomist with expertise in human and non-human primate brain microscopy, synaptic tract-tracing, and ultrastructure and I cannot judge the conditioning of the data and the statistical procedures used here by the authors, but I do acknowledge that the topic addressed in the present manuscript (the relation of molecular, structural, and functional levels in the human cerebral cortex) is quite interesting for a broad audience of both basic and clinical neuroscientists. Also, I do appreciate the experimental approach of combining different sets of data as productive and convenient. But I do have several serious conceptual concerns that pertain to the misuse of brain terminology, the originality of the identified principles, and the neglect of significant portions of scientific literature on the structure and connections of the human and non-human primate cerebral cortex.

1.- Proper use of neuroanatomical terminology: In the title the authors mention the cerebral cortex, but in the abstract and in most places through the text they talk about the brain. According to the Merriam-Webster dictionary, Brain is "the portion of the vertebrate central nervous system enclosed in the skull and continuous with the spinal cord", but it is obvious that the authors have not run data analysis of the thalamus or the striatum: they have analyzed the cerebral cortex. Please, note that the brain is not the cerebral cortex. Brain and cerebral cortex are not interchangeable terms.

2.- In search of principles and blueprints: In the abstract, the authors ambitiously declare that have "uncover a compact set of universal organizational principles whereby all types of inter-regional relationships reflect brain geometry and anatomical connectivity". Also, they state that their work sets nothing less than "the stage for next-generation connectomics and the integrative study of inter-regional relationships". These affirmations by the authors are hyperbolic because, as I will explain below, I could hardly find a single principle through the manuscript that had not been identified before using different techniques.

The major principle that runs through the manuscript is the relation between the laminar architecture of the cerebral cortex and its connections. In the past ten years there has been an explosion of human brain imaging articles in which the structural connectivity between cortical areas was related to "cortical microstructure", "Microscale pattern of cortical organization", "Microanatomy", "microcircuit specialization", "microscale properties", etc. The authors of most of these articles claim to be original in the identification of this principle (the relation between cortical structure and connections), but it was discovered in the 80s by Helen Barbas with synaptic tract-tracing in macaques. Basically, the human and non-human primate cerebral cortex shows a gradient of laminar complexity that is related to the laminar pattern (Barbas 1986; Barbas and Rempel-Clower 1997; García-Cabezas et al. 2019) and the strength (Aparicio-Rodríguez and García-Cabezas 2023) of cortico-cortical connections in a relational model called the Structural Model for Connections. This model allows for predicting the position of cortical areas across cortical hierarchies (Hilgetag and Goulas 2020) and has been used in in vivo MRI human cortex studies (Zhang et al. 2020). Also, it is related to the systematic variation of markers of synaptic plasticity and neuron stability (García-Cabezas et al. 2017; Zikopoulos et al. 2018). Even more, the Structural Model for Connections is compatible with causal mechanism of embryonic development and phylogeny (García-Cabezas et al. 2019; Puelles et al. 2019; García-Cabezas et al. 2022), a fundamental requisite to assert the biological meaning of any proposed principle. 

I think that the authors should carefully read the already classic literature of the Structural Model to realize that their findings are just a confirmation of principles identified long ago. The authors find that "across all seven connectivity modes, brain regions that are physically connected by white matter show greater feature similarity than those that are not connected, suggesting that biologically similar neuronal populations are in direct communication" and then claim to have found "that connectivity modes demonstrate common organizational principles that respect geometry, neuroanatomy, and anatomical connectivity, regardless of imaging modality or biological mechanism". In fact, the connectivity modes of the present manuscript just confirm the "organizational principles" that we already knew from neuroanatomical research in human and non-human primates and do not increase our insight on the biological mechanisms underlying the relation between cortical structure and cortico-cortical connections.

There are other places in the text were the authors claim to have identified a principle and try to relate it to biology. For instance, the authors find that the "edges in the brain's topological rich club regime are particularly dominated by molecular features (e.g. laminar similarity, correlated gene expression, and receptor similarity" and conclude that these results point to the possible biological origins of the rich club. Namely, the rich club may reflect coordinated patterns of inter-regional microscale similarity". The affirmation that these results point at the possible biological origins of the rich club or of any other structural or functional feature of the cerebral cortex needs a (at least partial) causal explanation, otherwise what the authors have found is not a principle but mere correlation without clear biological meaning. And in biology, causal mechanisms must be searched for in ontogeny and phylogeny.

Finally, at the end of the Discussion, the authors affirm that "The consistent primacy of molecular connectivity modes demonstrates that mapping brain connectivity from the perspective of the underlying biology—gene transcription,

receptor density, cellular composition—is just as, if not more, informative than oft-studied dynamical modes such as haemodynamic connectivity". So underlying biology matters in human cortex neuroscience. Is this a surprising conclusion?

3.- Neglect of classic scientific literature: At several places the authors seem to ignore relevant and fundamental pieces of scientific literature. It is like if they only knew about contemporary brain imaging articles; but contemporary neuroscience resulted from the integration of multiple approaches in multiple species. Neuroscientists of the XXIst century must be specialist in a given field of brain research but they should also be aware of fundamental knowledge from other fields.

For example, a significant finding in the present manuscript is the relation between molecular connectivity modes and selective vulnerability to neuropsychiatric disorders. The authors say that "Emerging theories emphasize that the course and expression of multiple brain diseases is mediated by shared molecular vulnerability [61, 169]". But this is far from being an emerging theory. Actually, Oskar and Cécile Vogt were the first in advance the concept of selective vulnerability and related it to differences in the physicochemical composition of neurons across brain regions. The authors can read the informed review of Klatzo (2003) to learn about the origin of selective vulnerability as concept.

Also, the authors affirm that "This suggests that brain regions with similar molecular makeup may undergo similar structural changes in disease". Again, the idea of different vulnerabilities across cortical regions in the human brain was already advanced and exemplified for Alzheimer´s disease by Braak and Braak (1996). 

Another example of classic and fundamental literature neglect is found in the Discussion where the authors state that "A second assumption, this stemming primarily from fMRI studies, is that the brain ubiquitously follows a unimodal-transmodal hierarchical gradient, and can be organized in terms of intrinsic resting-state networks". This is not emerging primarily form fMRI studies. On the contrary, it goes back to fundamental physiological and anatomical primate research done in the 60s, 70s, and 80s. I just recommend to the authors carefully reading the well-known article of Mesulam (1998).

4.- Neuroscience is not only about human brain imaging: I invite the authors to reconsider their statements about connectomics as the "increasingly becoming the dominant paradigm in neuroscience [13, 88, 145]". I invite them to explore the classical human and non-human neuroanatomy and neurophysiology literature, but also other contemporary paradigms, like those emanating from the developmental biology of the brain [e. g., Nieuwenhuys and Puelles (2016); Nieuwenhuys (2017)], that are changing our views of the human brain in many ways that are much deeper than in vivo brain imaging.

Miguel Ángel García-Cabezas.

References

Aparicio-Rodríguez G, García-Cabezas MÁ (2023) Comparison of the predictive power of two models of cortico-cortical connections in primates: The Distance Rule Model and the Structural Model. Cereb Cortex Online ahead of print. doi:doi: 10.1093/cercor/bhad104

Barbas H (1986) Pattern in the laminar origin of corticocortical connections. J Comp Neurol 252 (3):415-422. doi:10.1002/cne.902520310

Barbas H, Rempel-Clower N (1997) Cortical structure predicts the pattern of corticocortical connections. Cereb Cortex 7 (7):635-646

Braak H, Braak E (1996) Development of Alzheimer-related neurofibrillary changes in the neocortex inversely recapitulates cortical myelogenesis. Acta Neuropathol 92 (2):197-201. doi:10.1007/s004010050508

García-Cabezas MA, Hacker JL, Zikopoulos B (2022) Homology of neocortical areas in rats and primates based on cortical type analysis: an update of the Hypothesis on the Dual Origin of the Neocortex. Brain structure & function Online ahead of print. doi:doi.org/10.1007/s00429-022-02548-0

García-Cabezas MA, Joyce MKP, John YJ, Zikopoulos B, Barbas H (2017) Mirror trends of plasticity and stability indicators in primate prefrontal cortex. Eur J Neurosci 46 (8):2392-2405. doi:10.1111/ejn.13706

García-Cabezas MA, Zikopoulos B, Barbas H (2019) The Structural Model: A theory linking connections, plasticity, pathology, development and evolution of the cerebral cortex. Brain structure & function 224 (3):985-1008. doi:10.1007/s00429-019-01841-9

Hilgetag CC, Goulas A (2020) 'Hierarchy' in the organization of brain networks. Philos Trans R Soc Lond B Biol Sci 375 (1796):20190319. doi:10.1098/rstb.2019.0319

Klatzo I (2003) Cecile & Oskar Vogt: the significance of their contributions in modern neuroscience. Acta Neurochir Suppl 86:29-32. doi:10.1007/978-3-7091-0651-8_6

Mesulam MM (1998) From sensation to cognition. Brain 121:1013-1052

Nieuwenhuys R (2017) Principles of Current Vertebrate Neuromorphology. Brain Behav Evol 90 (2):117-130. doi:10.1159/000460237

Nieuwenhuys R, Puelles L (2016) Towards a New Neuromorphology. doi:10.1007/978-3-319-25693-1

Puelles L, Alonso A, Garcia-Calero E, Martinez-de-la-Torre M (2019) Concentric ring topology of mammalian cortical sectors and relevance for patterning studies. J Comp Neurol 527 (10):1731-1752. doi:10.1002/cne.24650

Zhang J, Scholtens LH, Wei Y, van den Heuvel MP, Chanes L, Barrett LF (2020) Topography impacts topology: Anatomically central areas exhibit a "High-Level Connector" profile in the human cortex. Cereb Cortex 30 (3):1357-1365. doi:10.1093/cercor/bhz171

Zikopoulos B, García-Cabezas MA, Barbas H (2018) Parallel trends in cortical gray and white matter architecture and connections in primates allow fine study of pathways in humans and reveal network disruptions in autism. PLoS Biol 16 (2):e2004559. doi:10.1371/journal.pbio.2004559

Reviewer #2 Basilis Zikopoulos (note, Reviewer 2 has signed this review): The paper entitled "Integrating multimodal, multiscale connectivity blueprints of the cerebral cortex" is a first effort to systematically compare and bring together distinct types of inter-regional similarity estimates that can and have been used to describe the organization of the cerebral cortex and its connectome. The findings suggest that all types of similarity estimates reflect more or less inter-regional topology and connectivity, which emerge as universal organizational principles of the cortex. Importantly, the authors also found unique relationships that emerge from the comparison of distinct types of inter-regional similarity estimates and highlight distinct associations between them. To this point, the authors highlight the variability of hubness profiles across modalities, which points out the importance of characterizing network architecture from multiple complementary perspectives. 

This is a very timely, interesting, and well written study and I want to commend the authors on their effort, care, and consideration of relevant literature, while completing this much-needed work. The strengths of this manuscript include (1) the impressive integration of seven multimodal and multiscale types of inter-regional cortical similarity, (2) the rigorous approach used to analyze and compare these seven modes, including sensitivity and replication analysis using multiple parcellation schemes, and (3) the development of a framework that can be used to correlate data from disparate sets of variables at multiple scales. 

The main limitations arise from compiling data for each modality across age, sex, and other key factors, like handedness (not mentioned in this article), limitations of each dataset used e.g., non-specific binding in PET datasets, and challenges when comparing inherently different types of data. Most were addressed adequately by the authors. As the authors stated, all limitations can be addressed in future in-depth studies, as new datasets become available for each feature used to determine these inter-regional similarity estimates however, I would clarify that they can also be addressed by concurrently using multiple datasets for each type of similarity estimate that are currently available. The authors addressed transmodal comparison challenges through an elegant study design that included alternative parcellation schemes, rank-transformations, and normalization of data. Therefore, despite the limitations, the findings are novel, highly significant, and the manuscript is poised to be an outstanding contribution to our field and of general interest. 

Below few comments and suggestions that in my opinion will further increase the value and clarity of the manuscript, improve presentation, and strengthen reported findings and discussion: 

In the introduction (first paragraph), the authors mention the prevalence of short-range connections that result in functionally segregated modules, as a hallmark feature of the structural architecture of the brain. A recent elegant study by Rosen and Halgren, PLOS Biology 2022 provided key evidence for the sparse connectivity in the human brain and should be cited here. 

The sentence starting the second paragraph is not entirely accurate and can be misleading leading to gaps in the presentation of the relationship between structure and function of the cortex presented in the second and third paragraphs. Several studies have shown that structure can account for key molecular and functional attributes of the brain, especially in the cortex. For example, Goulas et al., PNAS 2021 showed a receptor-based natural axis of the human cerebral cortex, which also parallels laminar architecture; Garcia-Cabezas et al., European Journal Of Neuroscience 2017 showed that cortical structure is systematically associated with several plasticity and stability markers; Garcia-Cabezas et al., Brain Structure and Function 2022 and Goulas et al., PLOS Biology 2019 showed how the organization of cortical structure and connectivity are rooted in the development and can be seen throughout evolution. 

The first part of the second paragraph of the introduction introduces key principles of the organization of the cortex including the fact that the most common connections are between similar areas, something that is true for primates and likely all mammals, extensively presented and discussed in several studies e.g., Garcia-Cabezas et al., Brain Structure and Function 2019. 

The use of the gwMRF parcellation of the cortex into 400 regions for the main analysis is first mentioned in the results and then in several other segments of the paper however, it is not clear why this parcellation scheme was selected for the main, detailed analysis over other commonly used schemes. More information on this and potential confounding effects based of the parcellation schemes used would be useful. Since there is little agreement between several, if not most, parcellation schemes, one could ask if the approach used in this paper could form the basis for a more general/universal parcellation of the cortex. 

In Results, page 2 the authors mention "Homotopic connections stand out, indicating that homologous

brain regions in left and right hemispheres are consistently similar to each other no matter the biological

feature". Relevant to this and other statements about connections involving one or both hemispheres, previous work in non-human primates has linked structure and connections in the cortex and showed parallel organization of contralateral and ipsilateral cortical projections in monkeys (see Barbas et al., BMC Neuroscience 2005). 

Page 2, Results Common organizational principles of connectivity modes, paragraph 2: "Furthermore, similarity between brain regions decreases as both Euclidean and geodesic distance between brain regions increases (Fig. 1b; Fig. S2), consistent with the notion that proximal neural elements are more similar to one another [47, 60, 67, 120, 137]" and later in Page 3, "These differences are greater than in a population of degree- and edge length-preserving surrogate structural connectomes, indicating that the effect is specifically due to wiring rather than spatial proximity [21]" and the section "Structural and geometric features of connectivity modes" the authors could place the current findings and their interpretation within the framework of prior work after reading and citing as needed relevant studies that have made similar points by e.g., Garcia-Cabezas et al., Brain Structure and Function 2019; Aparicio-Rodríguez and García-Cabezas, Cerebral Cortex 2023; Beul et al., Brain Structure and Function 2015; Beul et al., Scientific Reports 2017; Hilgetag et al., Network Neuroscience 2019. 

In Results, Cross-modal hubs, towards the end of the section (page 8) the authors state "We find that transmodal regions such as the supramarginal gyrus, superior parietal cortex, precuneus, and dorsolateral prefrontal cortex are most consistently similar to other brain regions across all connectivity modes (Fig. 3d, right). Interestingly, these transmodal regions are commonly thought of as structural hubs but here we show that they are central at multiple levels of organization. Why are some brain regions highly similar to many other regions across multiple spatial scales and biological mechanisms? We hypothesized that cross-modal hubs are more cognitively flexible and able to support higher order, evolutionarily-advanced cognitive processes. We therefore correlated cross-modal hubness with a map of evolutionary cortical expansion [68]". It would be beneficial to readers if the authors made a connection here with previous studies that have shown that these areas are similar in structure "eulaminate" regions that have significantly expanded in evolution in primates, especially humans (see Garcia-Cabezas et al., Brain Structure and Function 2022). 

The statement "transmodal regions are commonly thought of as structural hubs but here we show that they are central at multiple levels of organization" could be clarified further. Do the authors imply that all connectomes, including the structural, identify key hub areas that are more or less similar (contain the same areas) or not? Or is it that hubs across modalities tend to contain similar types of areas? Moreover, other studies (e.g., Zhang J, Scholtens LH, et al., Cerebral Cortex 2020) showed that highly anatomically central areas may function as "high-level connectors," integrating already highly integrated information across modules. These results are consistent with a high-level, domain-general limbic workspace, integrated by highly anatomically central cortical areas. How does that fit to the current findings for each distinct connectivity mode tested? 

I would appreciate more information about the relationship of exposure and negative edge strength in the Results "Connectivity modes shape disease vulnerability". Does a negative edge strength play any role in exposure calculation? Could a negative edge strength indicate resilience? 

It would be interesting to further examine how the findings from the current study fit/compare with relevant prior work with supporting or contrasting previous findings. For example, an elegant study by Goulas et al., PNAS 2021 uncovered a receptor-based natural axis of the human cerebral cortex, which also parallels laminar architecture and showed that on the sensory extreme of cortical gradients of laminar elaboration there is less diversity of receptors, more inhibition, less excitation, more ionotropic and fewer metabotropic receptors. On the association extreme of cortical gradients of laminar elaboration there is more diversity of receptors, less inhibition, more excitation, fewer ionotropic and more metabotropic receptors. I believe it would be interesting to add a comment about these observations and compare associations made in the two studies, especially given the following statement in page 9 "Furthermore, we find that brain gradients do not all follow a uniform sensory association axis [73, 96, 149], rather, the first principal component of each connectivity mode varies considerably". 

In addition, several other studies, reviewed in Garcia-Cabezas et al., Brain Structure and Function 2019 have linked structure of the cortex with presence/absence, strength, and laminar pattern of connections and function in terms of feedforward-feedback-lateral processing in the cortex. Again, it would be interesting to add a comment about how the findings from the current study fit/compare with these data. 

In "Connectivity modes shape disease vulnerability" and associated Fig. 4, the authors show how we can use this approach to hone in the imaging modalities and biological mechanisms that might most reflect cortical pathology. Later in the presentation of the results of the fusion of connectivity modes, the authors state that the fused network maps onto intrinsic networks and cytoarchitectonic classes better than any individual network, demonstrating how large-scale phenomena can emerge from a confluence of multiple determinants. Several diseases like ASD could be described as complex, large-scale phenomena, therefore it would be interesting to see whether the fused network could also map disease vulnerability better than any individual network and add this to Fig. 4. 

The structural connectivity dataset used may over-represent long-range cortical connections, which tend to be more common between eulaminate areas. The authors clarify that each dataset has its own limitations however, it seems that actual connectivity between areas (based on the structural connectivity dataset) weighs more during integration of multiple modes and within each mode for hub identification. The authors should discuss how this could affect cross-modal hub identification and other relevant findings. 

Reviewer #3: In this work, Hansen and colleagues explored different facets of brain connectivity, using measures of gene expression, receptor density, cellular composition, metabolism, electrophysiology, and temporal features. Specifically, they estimated a "connectivity mode" for each of these measures and investigated their common organizational principles and contributions to brain structure and geometry, also linking these measures with spatial patterns of cortical abnormality in a wide range of brain disorders. 

Overall, this is a very interesting piece of work, and finds its own space in the series of works from this group related to the topic of brain connectomics. The manuscript is well written, but I'd like to ask the authors to address some points before endorsing it for publication.

In section "Connectivity modes shape disease vulnerability", the authors say: "We next ask how connectivity modes shape the spatial patterning of brain diseases". Although I get what the authors mean, I'm not sure that the connectivity modes can 'shape' the patterning of brain diseases, nor that this can be tested in this work, as the only data of clinical cohorts used here are measures of structural abnormality, so not directly related to the modes investigated. Also, the authors seem to suggest that structural abnormalities define the spatial patterning of brain diseases, but the spatial patterning of brain diseases is much complex than this, and that the anatomical modifications are just part of the full picture. Some lines below, the authors use the term "cortical profile of the disease". This definition sounds more in line with what the authors probably want to indicate, although I would highlight that it's the structural profile of the disease.

In the same section, the authors say: "Finally, we correlate cortical abnormality with disease exposure to determine whether the spatial patterning of the disease is informed by a connectivity mode". I'm not quite convinced about the meaning of this sentence. What the authors are doing here is a correlation between structural alterations and connectivity modes, which means that they are checking if there is a link between the connectivity modes in the healthy brain and abnormal cortical thickness. The wording "is informed by" gives me the impression that, for example, if a correlation with the receptor mode exists, this mode is influencing the anatomical abnormality, but this is not necessarily the case and cannot be tested in this specific study. 

Same section: the authors should mention in this section that what they mean by "cortical abnormality" is "abnormal cortical thickness", as the term used is too generic.

Section "Gradients and modules of connectivity nodes" would benefit more details. For example:

* To give the readers a bit of context, I would provide a definition of gradients and modules and explain why they could be important to understand the connectivity modes.

* About the modules, I would find useful for non-expert readers an explanation of what the resolution parameter is, and what it means in practical terms that the solution is stable/unstable to define and understand the network's characteristics.

* "Collectively, this shows that each connectivity mode has a unique gradient decomposition and community structure." What does it mean in practical terms? Could you further explain why this information is relevant?

Section "Fusing connectivity modes":

* Why are the edge weights so small?

* Referring to the greater correlation between edge weight and weighted structural connectivity, the authors say: "This shows how combining inter-regional similarity across multiple scales can be used to better explain anatomical connectivity". I'm not sure about which measure can explain the other. Can this result be interpreted instead as the fact that anatomically connected regions are more likely to share similar profiles on multiple scales? So, basically, that it's the anatomical connectivity that can be used to better explain inter-region similarity across multiple scales?

* It would be interesting to localise in the brain map the regions/edges of the fused network, instead of having only a connectivity matrix, possibly trying to give an explanation of why regions A, B and C and their links are the core of this network.

Minor comments:

* I'd say that the definition of brain connectivity in the first line of the introduction refers more to the anatomical connectivity than other types of connectivity (e.g. functional connectivity).

* In the sentence "An emerging representation of connectivity is feature similarity: if two brain regions exhibit similar biological features, we might expect them to be related to one another and engaged in common function" can you provide a reference for this definition?

* While the meaning of brain structure is straightforward, the one of "brain geometry" has been used and misused in the literature, so I think the readers would benefit from an accurate definition of this concept.

* Is it 79800 the number of possible edges, given the chosen brain parcellation? If so, please specify it.

* Some sections are named with the type of measure extracted or analysis performed, e.g., "Cross-modal hubs" and "Fusing connectivity modes", others with the message that the authors want to convey with that section, e.g. "Connectivity modes shape disease vulnerability". I think that a unique approach to define each section would be easier to follow, and I find the latter more useful than the former.

* The last sentence of section "Connectivity modes shape disease vulnerability" is: "This integrative analysis makes it possible to hone in on the imaging modalities and biological mechanisms that might most reflect cortical pathology; in this case, bringing to light the relevance of molecular rather than dynamic modes in psychiatric disorders." It's not unexpected or a new concept that the molecular layer of the brain is relevant for the study of psychiatric disorders. I would change "bringing to light" with "confirming".

* In the electrophysiological connectivity section in the methods, I'd find more intuitive to have the description of the processing steps before the part related to the estimation of the connectivity matrix.

---

## [Decision Letter · Decision Letter 2]

15 Aug 2023

Dear Dr Hansen,

Thank you for your patience while we considered your revised manuscript "Integrating multimodal, multiscale connectivity blueprints of the cerebral cortex" for publication as a Research Article at PLOS Biology. This revised version of your manuscript has been evaluated by the PLOS Biology editors, the Academic Editor and by two of the original reviewers.

The reviewers and our Academic Editor think that you have done a good job addressing their concerns, and they are satisfied by the revision. However, before we can editorially accept your manuscript for publication, we need you to address a number of editorial requests, detailed below. 

**EDITORIAL REQUESTS: 

1) TITLE: We think that the title should be edited to indicate that human data was analyzed here and perhaps to capture the scope of the study in more detail. If you agree, we suggest it be changed to something like: 

"Integrated multimodal and multiscale connectivity blueprints of the human cerebral cortex in health and disease"

2) ABSTRACT: Similarly, the abstract should be edited to make clear and highlight that you are analyzing human data. 

3) BLURB: In the relevant section of our online system, please provide a blurb which (if accepted) will be included in our weekly and monthly Electronic Table of Contents, sent out to readers of PLOS Biology, and may be used to promote your article in social media. The blurb should be about 30-40 words long and is subject to editorial changes. It should, without exaggeration, entice people to read your manuscript. It should not be redundant with the title and should not contain acronyms or abbreviations.

4) DATA AVAILABILITY: Thank you for providing the code and data related to your paper on github. Can you please generate a DOI for this dataset? This can be done with zenodo : https://docs.github.com/en/repositories/archiving-a-github-repository/referencing-and-citing-content

Please also add a brief sentence to each figure legend (including supplemental) referencing this dataset. For example, you could add the sentence "The data underlying this figure can be found at https://github.com/netneurolab/hansen_many_networks

5) SUGGESTION FROM THE ACADEMIC EDITOR: Given that the authors were stating that previous studies in this field were “limited to qualitative measurements of cytoarchitectonic similarity, small subsets of brain regions “, they may also want to add the following paper to their reference list, wherein a group attempted to identify connectivity principles based on quantitative structural data for many primate cortical areas, although without having access to all the wonderful, multidimernsional comprehensive data that are now available: Hilgetag CC, Medalla M, Beul SF, Barbas H (2016) The primate connectome in context: Principles of connections of the cortical visual system. Neuroimage 134:685-702. doi: 10.1016/j.neuroimage.2016.04.017

We expect to receive your revised manuscript within two weeks. 

*Published Peer Review History*

*Press*

Sincerely,

Luke

Lucas Smith, Ph.D.

Senior Editor,

lsmith@plos.org,

PLOS Biology

Reviewer remarks:

Reviewer #1 Miguel Angel Garcia Cabezas (reviewer 1 has signed this review): The authors have addressed my concerns.

Reviewer #2 Basilis Zikopoulos (reviewer 2 has signed this review): The authors did an excellent job addressing all my comments, clarified important concepts, performed additional analyses, highlighted the importance of this work, and set the stage for exciting future studies. Well done.

---

## [Editor Report · Decision Letter 3]

28 Aug 2023

Dear Dr Hansen,

Thank you for the submission of your revised Research Article "Integrating multimodal and multiscale connectivity blueprints of the human cerebral cortex in health and disease" for publication in PLOS Biology, and thank you for addressing our last editorial requests in this revision. On behalf of my colleagues and the Academic Editor, Claus C. Hilgetag, I am pleased to say that we can in principle accept your manuscript for publication, provided you address any remaining formatting and reporting issues. These will be detailed in an email you should receive within 2-3 business days from our colleagues in the journal operations team; no action is required from you until then. Please note that we will not be able to formally accept your manuscript and schedule it for publication until you have completed any requested changes.

PRESS

Sincerely, 

Lucas Smith, Ph.D.

Senior Editor

PLOS Biology

lsmith@plos.org